



# Revealing the significant acceleration of Hydrofluorocarbon (HFCs) emissions in eastern Asia through long-term atmospheric observations

Haklim Choi[1], Alison L. Redington[2], Hyeri Park[3], Jooil Kim[4], Rona L. Thompson[5], Jens Mühle[4], Peter K. Salameh[4], Christina M. Harth[4], Ray F. Weiss[4], Alistair J. Manning[2], and Sunyoung Park[1,3,*]

[1]Kyungpook Institute of Oceanography, Kyungpook National University, Daegu, Republic of Korea
[2]Hadley Centre, Met Office, Exeter, UK
[3]Department of Oceanography, Kyungpook National University, Daegu, Republic of Korea
[4]Scripps Institution of Oceanography, University of California San Diego, La Jolla, California, USA
[5]NILU – Norsk Institutt for Luftforskning, Kjeller, Norway

*Correspondence to*: Sunyoung Park (sparky@knu.ac.kr)

**Abstract.** Hydrofluorocarbons (HFCs) are powerful anthropogenic greenhouse gases (GHGs) with high global warming potentials (GWPs). They have been widely used as refrigerants, insulation foam blowing agents, aerosol propellants, and fire suppression agents. Since the mid-1990s, emissions of HFCs have been increasing rapidly as they are used in many applications to replace ozone depleting chlorofluorocarbons (CFCs) and hydrochlorofluorocarbons (HCFCs) whose consumption and production have been phased out under the Montreal Protocol (MP). Due to the high GWP of HFCs, the Kigali amendment to the MP requires the phase-down of production and consumption of HFCs to gradually achieve an 80–85% reduction by 2047 starting in 2019 for non-Article 5 (developed) countries with a 10% reduction against each defined baseline and later schedules for Article 5 (developing) countries. In this study, we have examined long-term high precision measurements of atmospheric abundances of 5 major HFCs (HFC-134a, HFC-143a, HFC-32, HFC-125, and HFC-152a) at Gosan station, Jeju Island, South Korea from 2008 to 2020. Background abundances of HFCs gradually increased, and the inflow of polluted air masses with elevated abundances from surrounding source regions were detected over the entire period. From these pollution events, we inferred regional and country-specific HFC emission estimates using two independent Lagrangian particle dispersion models and Bayesian inversion frameworks (FLEXPART-FLEXINVERT+ and NAME-InTEM). The spatial distribution of the derived "top-down" (measurement based) emissions for all HFCs shows large fluxes from megacities and industrial areas in the region. Our most important finding is that HFC emissions in eastern China and Japan have sharply increased since 2016. The contribution of East Asian HFC emissions to the global total increased from 9% (2008–2015) to 15% (2016–2020). In particular, HFCs emissions in Japan (Annex 1 country) rose rapidly from 2016 onward, with accumulated total inferred HFCs emissions being ~76 Gg/yr higher for 2016–2020 than the "bottom-up" (i.e., based on activity data and emission factors)



emissions reported to the United Nations Framework Convention on Climate Change (UNFCCC). This is likely related to the increase in domestic demand in Japan for refrigerants and air-conditioning system-related products and incomplete accounting. A downward trend of HFCs emissions that started in 2019 reflects the effectiveness of the F-gas policy in Japan. Eastern China and South Korea, though not obligated to report to UNFCCC, voluntarily reported emissions, which also show differences between top-down and bottom-up emission estimates, demonstrating the need for atmospheric measurements, comprehensive data analysis and accurate reporting for precise emissions management.

## 1 Introduction

Hydrofluorocarbons (HFCs) were introduced to replace stratospheric ozone-depleting substances (ODSs) such as chlorofluorocarbons (CFCs) and hydrochlorofluorocarbons (HCFCs) whose production and consumption are regulated under the Montreal Protocol (UNEP, 1987). HFCs are widely used in industrial applications such as air conditioning, refrigeration, foam blowing, and fire extinguishers. HFCs are not ODSs since they are oxidised more readily in the troposphere and do not contain chlorine atoms. However, HFCs are potent anthropogenic greenhouse gases (GHGs) with global warming potentials (GWPs) about hundreds to thousands of times higher than that of carbon dioxide ($CO_2$).

Due to their high GWP, HFCs have been included in the Kyoto Protocol, which aims to reduce greenhouse gas emissions (Breidenich et al., 1998). Additionally, nations in the Annex-1 group that have made the commitment to mitigate climate change (within their jurisdiction and capacity to do so) must provide yearly emissions statistics to the UNFCCC (United Nations Framework Convention on Climate Change). However, the Kyoto Protocol was not ratified by all countries (only Annex I countries), so it was superseded by the Paris Agreement (Paris agreement, 2015). The Kigali Amendment to the Montreal Protocol was adopted in 2015, with the goal of gradually reducing HFC production and consumption globally to attain an 80-85% decrease by 2047. Under the Amendment, Parties to the Montreal Protocol are divided into four groups (non-Article 5 earlier starts, non-Article 5 later starts, Article 5 Group 1, and Article 5 Group 2), and each sets a baseline for HFCs and proceeds with a phase-down under the Kigali Amendment. Non-Article 5 early starts and later starts countries begin phasing down in 2019 and 2020 in order to reach an 85% reduction in 2036. In addition, Article 5 Groups 1 and 2 Parties have reduction targets of 80% by 2045 and 85% by 2047, with reductions starting at 10% in 2029 and 2032, respectively. This phase down schedule for Article 5 countries is relatively lenient compared to non-Article 5 countries. Reliable HFCs emissions estimates are required to monitor the phase-down under the Kigali Amendment.

There are two ways to quantify HFCs emissions: 1) "bottom-up", where emissions are determined by applying emission factors based on each sector's production/consumption (activity data). These are reported by developed countries as national emission inventory (NEI) data to the UNFCCC. Some developing countries also compile NEI, but do not report to UNFCCC. 2) "top-down", where emissions are inferred from direct measurements of atmospheric abundances combined with inverse modeling or interspecies-correlation methods. Understanding these two complementary approaches and evaluating potential discrepancies between them are essential to reducing uncertainty in quantifying surface emissions on global and national scales.



Rigby et al., 2014 demonstrated that global HFCs emissions estimated from the AGAGE observation network and the national inventories reported to the UNFCCC from Annex-1 countries were in relatively good agreement until the early 1990s, but then the gap between the two estimates widened as global top-down emissions continued to increase, while the bottom-up emissions reported by Annex-1 countries plateaued after the late 1990s. In 2011, global top-down emissions were approximately two times the bottom-up emissions reported by Annex I to the UNFCCC. These discrepancies have persisted and the gaps are still growing (Velders et al., 2022). This gap may reflect differences in reported emissions from Annex I countries, as well as the impact of continued increases in emissions from non-Annex I countries.

To understand where global emissions arise, it is necessary to estimate emissions on national and regional scales (Weiss and Prinn, 2011). Previous regional studies conducted in Europe (Graziosi et al., 2017) and the United States (Hu et al., 2017) have demonstrated that there is reasonable consistency between bottom-up and top-down estimates of emissions. This suggests that the discrepancies at the global scale are less likely to be caused by significant under-reporting from Annex I countries, but rather arise from unreported emissions from developing countries, even though some gaps exist for specific HFCs in some Annex I countries (Lunt et al., 2015; Manning et al., 2021).

According to the statistical summary from Flerlage et al. (2021), many studies have been conducted worldwide to estimate regional-scale HFC emissions. Nevertheless, regions such as Latin America and the Caribbean (LAC), Africa, and India suffer from a dearth of observation stations and top-down estimation studies. In eastern Asia, which contributes significantly to global HFC emissions, top-down and bottom-up studies targeting the whole of China (Yao et al., 2019 and Fang et al., 2016) as well as specific regions in China such as the Yangtze River Delta (Pu et al., 2020), Greater Pearl River Delta (Zeng et al., 2020), North China Plain (Ding et al., 2023), and big cities (Yi et al., 2023) have been conducted. Until the early 2010s, many studies estimated HFC emissions for a limited period of time in the eastern Asia region (Japan, South Korea, North Korea, and Taiwan), excluding China (Fortems-Cheiney et al., 2015; Li et al., 2011; Lunt et al., 2015; Stohl et al., 2010). Since then, however, there has been a lack of recent research on HFCs, culminating in a dearth of information on current HFCs emission trends in eastern Asia. As the reduction of HFCs emission is gradually being achieved globally under the Kigali Amendment, it is necessary to continuously monitor and identify long-term HFCs emission trends in eastern Asia.

Therefore, we present a high-precision, high-frequency measurement record of 5 major HFCs (HFC-134a, HFC-32, HFC-125, HFC-143a, and HFC-152a) observed at Gosan station, Jeju Island, South Korea from 2008 to 2020, and analyze long-term temporal variations. We use the long-term observations to estimate regional top-down emissions at the country-level in eastern Asia using a Bayesian inversion approach. We compare these to bottom-up inventories reported to the UNFCCC and the Emission Database for Global Atmospheric Research (EDGAR) version 7 (Crippa et al., 2021) by each country to better understand recent HFCs emissions in eastern Asia and how they may be affected by the Kyoto protocol, the Paris agreement, and the Kigali amendment.





## 2 Data and Methodology

### 2.1 Instrumentation

Atmospheric abundances of HFCs were measured at Gosan (GSN, 33.3°N, 126.2°E, 72 m a.s.l) at the southwestern cliff of
Jeju Island, using a "Medusa" gas chromatographic system with cryogenic preconcentration and mass spectrometric detection (Miller et al., 2008; Arnold et al., 2012; Prinn et al., 2018). Gosan is an ideal location for determining regional emissions because of the minimal influence from local anthropogenic sources combined with inflow of strongly polluted air masses from around regions such as China, Japan, and Korea interspersed with periods of inflow of relatively unpolluted air masses from the Northern and Southern Hemisphere.

The Medusa system has been measuring more than 50 halogenated substances with high-precision, including the 5 major HFCs studied here, approximately every 2-hours (12 times a day) since 2008 as a part of the Advanced Global Atmospheric Gases Experiment (AGAGE; Prinn et al., 2018).

The observed atmospheric abundances (mole fractions) of HFCs are reported on calibration scales maintained by Scripps Institution of Oceanography (SIO, SIO-05 for HFC-134a and HFC-152a, SIO-07 for HFC-32 and HFC-143a, and SIO-14 for
HFC-125). The measurement precision ($1\sigma$) for HFCs species determined through repeated analyses ($n$=12) of ambient working standards is better than 2% (i.e., precision for HFC-134a < 1%, HFC-32 < 5% (improving from ~5% in 2008 to < 2% by 2017), HFC-125 < 1%, HFC-143a < 2%, HFC-152a < 2%).

**Table 1: Detailed information, lifetime, molecular weight, radiative efficiency, 100-year global warming potential, and applications**
**(Liang and Rigby et al. (2022), for the 5 major hydrofluorocarbons HFC-134a, HFC-143a, HFC-32, HFC-125, and HFC-152a.**

| Industrial Name | Chemical Formula | Lifetime (year) | Molecular weight (g mol$^{-1}$) | Radiative Efficiency (Wm$^{-2}$ppb$^{-1}$) | GWP$_{100}$ | Main applications |
|---|---|---|---|---|---|---|
| HFC-134a | $CH_2FCF_3$ | 14 | 102.0 | 0.16 | 1470 | mobile air conditioning refrigerant refrigerant blend component metered-dose inhalers aerosol propellant |
| HFC-143a | $CH_3CF_3$ | 51 | 84.0 | 0.16 | 5900 | refrigerant blend component |
| HFC-32 | $CH_2F_2$ | 5.4 | 52.0 | 0.11 | 749 | refrigerant blend component refrigerant |
| HFC-125 | $CHF_2CF_3$ | 30 | 120.0 | 0.23 | 3820 | refrigerant blend component fire suppression |



| | | | | | | |
|---|---|---|---|---|---|---|
| HFC-152a | CH$_3$CHF$_2$ | 1.6 | 66.1 | 0.10 | 153 | foam-blowing agent<br>aerosol propellant |

## 2.2 Inversion frameworks for estimating regional HFC emissions

Several Bayesian inversion models based on Lagrangian particle dispersion models (LPDMs) have been developed to estimate regional or national-scale emissions from high-precision atmospheric measurements, and many studies have quantified annual emissions of trace gases through intercomparison of inversion methods (Rigby et al., 2019; Park et al., 2021). Therefore, to

get a better understanding of annual HFC emissions, their uncertainties, and variability inferred from observation at Gosan, we have used two fully independent Bayesian inversion frameworks, FLEXINVERT+ (Thompson and Stohl, 2014) and Inversion Technique for Emission Modelling (InTEM; Manning et al., 2011) coupled with FLEXible PARTicle dispersion model (FLEXPART; Pisso et al., 2019) and Numerical Atmospheric dispersion Modelling Environment (NAME; Jones et al., 2007), respectively. The two inversion frameworks use different meteorological data and LPDMs, and their inversion methods

differ in terms of prior information and uncertainties, background concentrations, model domain, and other parameters (Table 2). Both frameworks can estimate trace gas emissions using a range of temporal resolutions. In this study, for both inversion frameworks we chose to use a 2-year assimilation time window, which increased the sensitivity through a more extended temporal window than compared to a 1-yr window (which is commonly used for synthetic gases), and to resolve the emissions with 1-year resolution. The longer assimilation window provides smoother, more realistic annual variation in emissions. The

emissions are derived using a 2-year inversion period advanced in steps of 1-year. The annual emissions shown are the average of the two inversions that include that year, with the exception of 2008 and 2020 which are based on a single 2-year inversion. Hereafter, the emissions and uncertainty ranges presented in the results of this study are each of mean values without weighting to represent the results derived independently from the two inversion frameworks. The configuration of each inversion frameworks is outlined in Table 2 and the description of each inversion framework are as follows.


**Table 2: The details of construction for two independent Bayesian inversion frameworks, FLEXPART-FLEXINVERT+ and NAME-InTEM.**

| | FLEXPART-FLEXIVNERT+ | NAME-InTEM |
|---|---|---|
| Lagrangian Particle Dispersion Model | FLEXPART v10.4 | NAME |
| Number of particles | 50,000 | 20,000 |
| Backward time | 20 days | 30 days |
| Meteorological data | NCEP CFSR | UK Met-Office UM |



| Horizontal resolution of meteorology | $0.5° \times 0.5°$ | $0.563° \times 0.375°$ to $0.141° \times 0.094°$ |
|---|---|---|
| Inversion period | 2 years | 2 years |
| Key references | Thompson and Stohl, 2014 Kim et al., 2021 | Manning et al., 2011 and 2021 Arnold et al., 2018 |

### 2.2.1 FLEXPART-FLEXINVERT+


A detailed description of FLEXPART-FLEXINVERT+, including the theoretical approach, and the presetting process applied in this study are given in Kim et al., 2021. Here, we present a brief overview of the inverse methods.

The origin of the air masses observed at 2-h intervals at Gosan and their path of diffusion were traced using FLEXPART v10.4 (Pisso et al., 2019), an LPDM, using the National Centers for Environmental Prediction (NCEP) Climate Forecast System

Reanalysis (CFSR; Saha et al., 2014) with a horizontal resolution of $0.5° \times 0.5°$ and a 1-h temporal resolution as input. To simulate the transport and dispersion of atmospheric particles, we released 50,000 particles at Gosan and tracked their movement backward in time for 20 days. The footprint sensitivities, a source-receptor relationship matrix, which indicates how spatially sensitive the observations at Gosan are to emissions at any given source location, were estimated at the same spatial resolution ($0.5° \times 0.5°$) as the input NCEP CFSR meteorological data.

FLEXINVERT+ is a Bayesian inversion model coupled with the FLEXPART model, which is designed to estimate surface fluxes from continuous observation time-series data. The Bayesian approach improves the accuracy of the estimated surface fluxes by accounting for uncertainties in the system. This method estimates the optimal posterior state within the model domain from the prior information and atmospheric observation time series with considering each uncertainty by minimizing the cost function as follows:


$$J(p) = \frac{1}{2}(p - p_0)^T \mathrm{B}^{-1}(p - p_0) + \frac{1}{2}(\mathrm{H}(p) - y)^T \mathrm{R}^{-1}(\mathrm{H}(p) - y) \tag{1}$$

where, $p$ and $p_0$ are the state vector and its prior information, $y$ is the measured enhancement that is determined as the elevated mole fraction over the background state. The regional background condition, which is called baseline, is determined by

separating the pollution signals according to a statistical classification method developed in AGAGE (O'Doherty et al., 2001; see Section 3.1 for details). B and R are prior and observation error covariance matrix, respectively. H is the atmospheric chemistry transport function (via continuity equation).

In applying the temporally observed data at Gosan to the inversion framework, we use all measured "enhancements" with an interval of about 2 hours, without temporal averaging. The total error ($\sigma_{tot}$) applied to each enhancement point is determined

by applying the error propagation method to the three components related to instrument, background, and model.



$$\sigma_{tot} = \sqrt{\sigma_{inst}^2 + \sigma_{bkg}^2 + \sigma_{model}^2} \qquad (2)$$

Where, $\sigma_{inst}$ is the instrumental precision of each HFC compound based on the repeatability measurement for the working

tank. $\sigma_{bkg}$ is the monthly standard deviation of background mole fraction which was determined to define the enhancement. The mean and standard deviation of the monthly background mole fractions were assumed to be at the midpoint of each month, the 15th, and then linearly interpolated for each observation point. $\sigma_{model}$ denotes the model representation error, which corresponds to a value of approximately 3% of the background mole fraction.

As shown in Equation 1 in a Bayesian approach, the spatial distribution and quantitative amount of prior information, and its

associated uncertainty can affect the final optimized posterior results. The spatial distribution of prior emissions of each HFC was spatially weighted by the global population density (CIESIN, 2015) based on global HFC emissions estimated for 2008 by Rigby et al. (2014). We established regional emissions for 2008 from previous studies (Stohl et al., 2010 for HFC-134a and Li et al., 2011 for other HFCs) for the highly sensitive regions of eastern Asia, such as eastern China, Japan, South Korea, North Korea, and Taiwan. The emissions allocated to individual regions for each HFC are summarized in Table S1. These

prior emissions defined for 2008 were applied equally to all years. To account for the uncertainty in initial emissions, we constructed an ensemble of emission distributions, magnitudes, and uncertainties comprising 27 datasets. These datasets were generated based on various conditions, including three types of initial emission distributions ("population distribution", "Asia flattened', and 'eastern Asia flattened"), three quantitatively scaled emission levels (×0.5, ×1, and ×2), and three initial uncertainty levels (100%, 200%, and 300%) (refer to Figure S1 for further details about spatial distribution).


**2.2.2 NAME-InTEM**

To ensure the consistency and accuracy of estimated emissions for each HFC, this study simultaneously utilizes NAME-InTEM, a well-established inversion model that has been widely applied in previous research to estimate emissions of various greenhouse gases. InTEM, a Bayesian inversion model with a non-negative least squares solver, uses the NAME Lagrangian

dispersion model developed by the UK Met Office to trace the movement and dispersion of air trajectories. In this study, NAME is employed to track the trajectory of air movement over the past 30 days by releasing 20,000 particles per hour from the location of Gosan station. Similar to FLEXPART-FLEXINVERT+, NAME-InTEM constructs a spatial distribution of HFCs prior emissions based on the population density (but not uniformly flattened for a certain region) and sets the prior uncertainty to 100%.

To exclude differences arising from the variability of observation sites, InTEM estimates emissions using only the same Gosan observation data. The inversion process, performs 24 iterations repeatedly for the same time period whilst removing 8 randomly selected blocks of 5 days within the temporal window to improve the estimates of uncertainty. For more detailed information regarding the model and methodology, see Arnold et al., 2018 and Manning et al., 2021.



## 3 Results

### 3.1 Long-term measurements of atmospheric HFCs at Gosan

Long-term, high-frequency atmospheric HFCs mole fractions observed during 2008–2020 at Gosan are illustrated in Figure 1. The regional background mole fraction of each HFC was determined by applying AGAGE's statistical algorithm via Gaussian filtering (O'Doherty et al., 2001) to separate pollution signals within a 121-days temporal window (± 60 days from each observation time).

The increasing trend of background mole fractions was continuous and evident for all HFCs. The annual average background mole fractions of HFC-134a and HFC-143a have linearly and continuously increased from 53.3 ± 3.0 ppt and 9.5 ± 0.5 ppt in 2008 to 120.5 ± 3.7 ppt and 27.3 ± 0.3 ppt in 2020, respectively. HFC-32 and HFC-125 have shown steeper, exponential increase in background mole fractions from 3.3 ± 0.2 ppt and 7.4 ± 0.5 ppt in 2008 to 29.1 ± 2.0 ppt and 35.7 ± 1.5 ppt in 2020, respectively. In contrast, HFC-152a showed a relatively moderate increase from 8.2 ± 1.5 ppt in 2008 to 10.0 ± 2.0 ppt in 2020, reflecting its short lifetime of 1.6 years. These increases in HFCs background mole fractions in eastern Asia are similar to the global trends presented in Montzka and Velders (2018) and Liang and Rigby et al. (2022).

For all HFCs, we observe a seasonal drop of background mole fractions due to the summer Asian monsoon bringing relatively clean air masses from the Tropical Pacific to Gosan (Li et al., 2018). Several gaps in the observations were caused by instrumental shut-downs due to typhoons or technical problems (Choi et al., 2022).

On top of the increasing HFC background mole fractions, pollution signals have been observed throughout the entire period and their magnitudes have also progressively increased as shown in Figure 2, which implies persistent, increasing HFC emissions from nearby anthropogenic sources in eastern Asia.





**Figure 1: Atmospheric HFCs mole fractions measured at Gosan from 2008 to 2020. The baseline data (black) are selected using the AGAGE statistical method (O'Doherty et al., 2001) to determine polluted data (red) which are elevated above the baseline data.**




**Figure 2: Box-whisker plots of annual mole fraction enhancements above baseline for each HFC from 2008 to 2020. The box encompasses the interquartile range (IQR), which is defined as the 25th–75th percentiles, while the whiskers represent the maximum and minimum enhancements. The solid lines indicate the median value of the data.**




## 3.2 Top-down emissions estimates of HFCs in eastern Asia



**Figure 3: Top-down estimates of annual HFCs emissions in eastern Asia. Annual emissions (solid lines) and uncertainty ranges (shading) are defined as the average of inversion posterior and 1-$\sigma$ uncertainty derived from FLEXPART-FLEXINVERT+ and NAME-INTEM. Each HFCs emissions result derived for individual models is shown in Figures S2 to S6.**



### 3.2.1 HFC-134a

HFC-134a is the most abundant HFC in the atmosphere. It has been mainly used since the mid-1990s as a refrigerant in Mobile Air Conditioner systems (MACs) primarily as a replacement for CFC-12, and is also used in inhalers, blend component for
stationary air-conditioning and commercial refrigeration, and as aerosol propellant (Montzka and Velders et al., 2018; Liang and Rigby et al., 2022).

As shown in Figure 3 and Table 3, HFC-134a emissions in eastern China presented the most dramatic growth in eastern Asia, with a slow increase from 3.2 ± 0.8 Gg/yr in 2008 to 6.6 ± 1.8 Gg/yr in 2014 (on average 5.0 ± 1.3 Gg/yr), and then a rapid rise after 2015 reaching a peak of 11.7 ± 2.9 Gg/yr in 2016. From 2016, the emissions plateaued, and from 2016 to 2018
remained relatively constant at an average of 11.2 ± 0.6 Gg/yr. Interestingly, in Japan, the annual HFC-134a emissions remained relatively constant at 6.5 ± 0.3 Gg/yr from 2008 to 2015, but showed a sharp increase after 2016 reaching 12.4 ± 2.5 Gg/yr in 2018, and then decreased since 2019. Until 2010, Japan had emitted more HFC-134a than eastern China, but since 2011, emissions from Japan and eastern China have become similar. While the emissions from South Korea, North Korea, and Taiwan are relatively small, they have also been steadily increasing. Due to the nearly negligible emissions from North Korea
and Taiwan, these regions are not discussed.

The rise in HFC-134a emissions in eastern Asia was attributed to the increased adoption of HFC-134a in response to economic and industrial growth in East Asian countries following the phase-out of CFC-12, whose production and consumption was banned by the Montreal Protocol. This is also consistent with a significant decline in CFC-12 emissions since the late 2000s in eastern Asia (Park et al. 2021).
The annual HFC-134a emissions estimates for each country are slightly different from the national inventories reported to the UNFCCC (data available at the UNFCCC greenhouse gas inventory website, https://di.unfccc.int, last access: 24 May 2023) and the EDGAR-v7 (Figure 4). China reported the national HFC-134a emissions to the UNFCCC for three years, 2010, 2012, and 2014, and the reported emissions had increased over the years. Comparing the reported emissions for the whole of China, downscaled by population density (35% for eastern China), to those estimated by the top-down method in this study, we find
that they were almost identical within the uncertainty range in 2010, but differences increased to 4.2 Gg/yr and 8.1 Gg/yr in 2012 and 2014, respectively (top-down emissions accounted for 58% and 44% of the reported inventories, respectively). Japan reported to UNFCCC that its HFC-134a emissions gradually and slowly decreased from ~2.8 Gg/yr in 2008 to ~2.3 Gg/yr in 2019. However, the inferred top-down emissions for 2008–2015 were almost twice as large, when annual emissions remained constant, and the discrepancies increased further with the sharp rise in the inferred emissions from 2016 onward. In contrast,
for South Korea, HFC-134a emissions estimated in the national inventory were on average 3.7 Gg/yr higher than the inferred top-down emissions over the entire period (top-down emissions were about 38% of the bottom-up inventories).





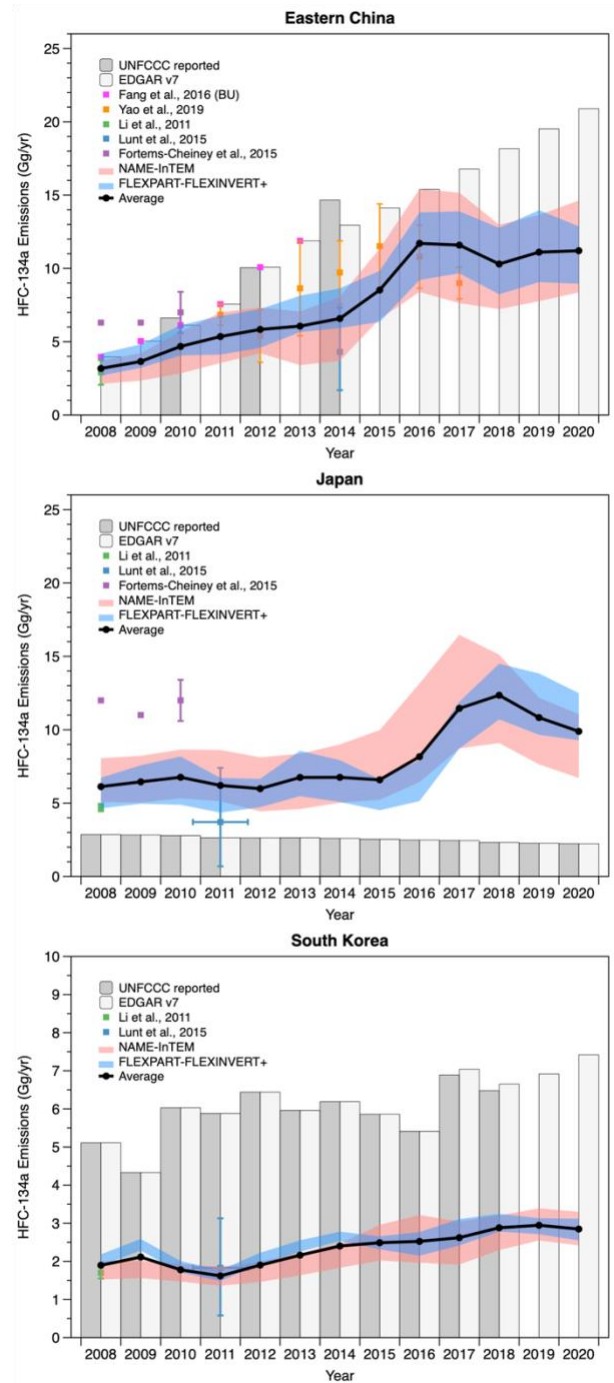

**Figure 4: The annual HFC-134a emissions for eastern China, Japan, and South Korea derived from FLEXPART-FLEXINVERT+ (in blue shaded) and NAME-InTEM (in red shaded), and average values (black solid line), respectively. Each country's HFC-134a emissions were compared with the inventories reported to the UNFCCC (dark gray bars), EDGAR v7 (light gray bars) and other studies. Note that, comparison group emissions for China have been downscaled relative to population density for eastern China from reported or derived emission for whole China.**






### 3.2.2 HFC-32

Until recently, HFC-32 was primarily used as a blend with HFC-125 (50% of HFC-32 and 50% of HFC-125) in the air
conditioning refrigerant R-410A, which is an alternative to the ozone-depleting HCFC-22 (R-22). HFC-32 (R-32) is also used
as a stand-alone refrigerant in commercial and residential air conditioners due to the GWP of one-third that of R-410A. HFC-
32 is one of the substances with an abundance that has been increasing substantially in recent years. HFC-32 emissions in
eastern China increased from 1.6 Gg/yr in 2008 to a peak of 10 Gg/yr in 2019, and slightly decreased to 9 Gg/yr in 2020.

280    Eastern China has been emitting the largest amount of HFC-32 in the study region. As shown in Figure 5, these annual HFC-
32 emissions are similar to those reported in EDGAR v7. In contrast, down-scaling the reported emissions for China for 2010,
2012, and 2014 to Eastern China using population density indicates much lower emissions of <1 Gg/yr), in stark contrast to
the significant emissions derived here. For Japan, the second highest emitter of HFC-32, top-down emissions gradually
increased from 0.4 to 1.9 Gg/yr between 2008 and 2015, with a peak of 5.3 Gg/yr in 2018, followed by a decline since 2019.

285    These top-down emissions match fairly well the reported inventory to the UNFCCC for the period of 2008–2015, but the
inventory does not show the rise in emissions during 2016–2019, but in 2020 there is a good match again.





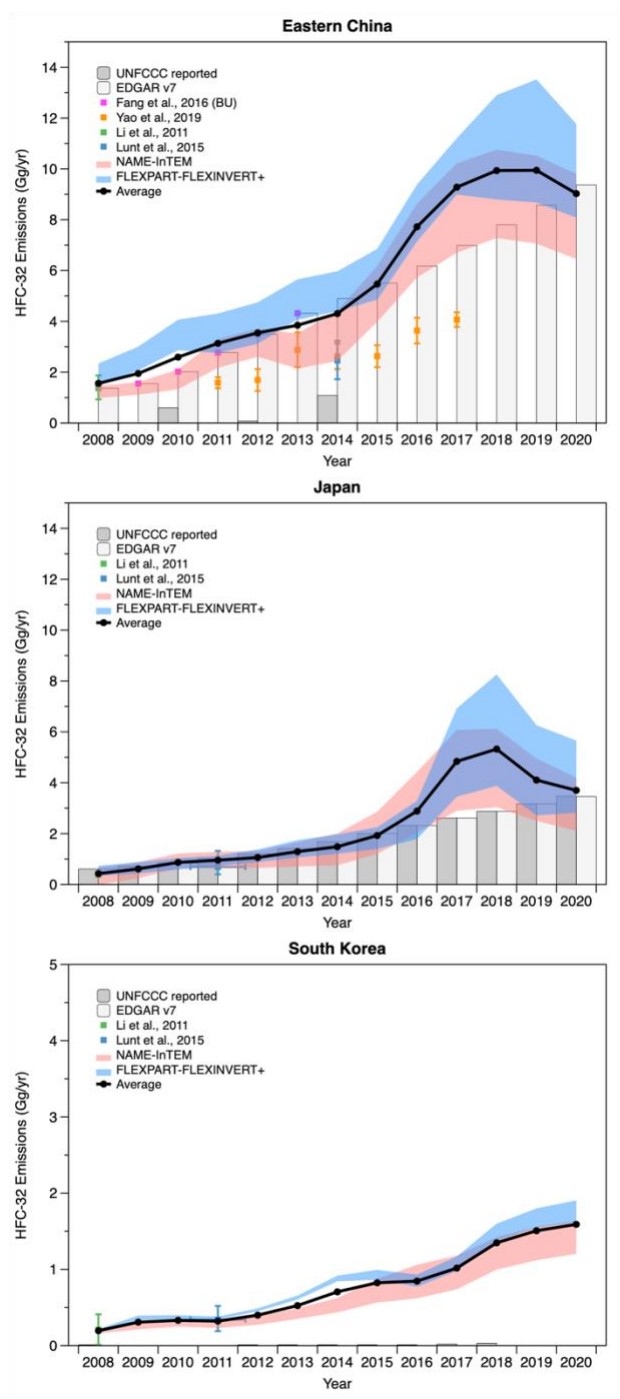

**Figure 5: Same as Fig. 4, but for HFC-32.**



### 3.2.3 HFC-125

As mentioned, HFC-125 is primarily used in refrigerant blends, foremost R-410A, a mixture with HFC-32. HFC-125 is also contained in other refrigerant blends, such as R-404A, R-407A, R-407C, R-407F, etc. HFC-125 emissions in eastern China increased rapidly from 1.2 Gg/yr in 2008 to a peak of 7.4 Gg/yr in 2019, with a slight decline to 7.1 Gg/yr in 2020. These inferred top-down emissions are much larger than the inventory emissions reported to UNFCCC for 2010, 2012, and 2014 (when down-scaled to eastern China), similar to the discrepancies for HFC-32, but top-down emissions are in good agreement with EDGAR-v7. Between 2008 and 2015, Japan's HFC-125 emissions showed a slight increase from 1.1 to 2.3 Gg/yr, followed by a rapid increase in the emissions reaching 5.1 Gg/yr in 2018, and decline thereafter. Comparing the emissions reported to UNFCCC with the top-down emissions inferred here (Figure 6), a close agreement within the uncertainty was observed between 2008 and 2015, similar to HFC-32. However, top-down emissions rose significantly above reported emissions during 2016-2018, with the maximum gap of 3.1 Gg/yr in 2018 when inferred emissions were ~2.5 times higher than reported emissions. The discrepancy declined somewhat in 2019 and 2020. South Korea showed a steady increase from 2008 to 2020 (0.3 to 1.8 Gg/yr). Emissions from North Korea were relatively constant from 2008–2015 with an average of $0.06 \pm 0.03$ Gg/yr, followed by an increase in 2016–2017 and then a slight decline to ~0.24 Gg/yr during 2018-2020. The emissions in Taiwan were almost negligible until 2012, but have been gradually increasing since 2013. The changes in annual HFC-125 emissions from year to year for each country were very similar to those of HFC-32 emissions, suggesting that emissions of HFC-32 and HFC-125 are primarily driven by the use of R-410A.



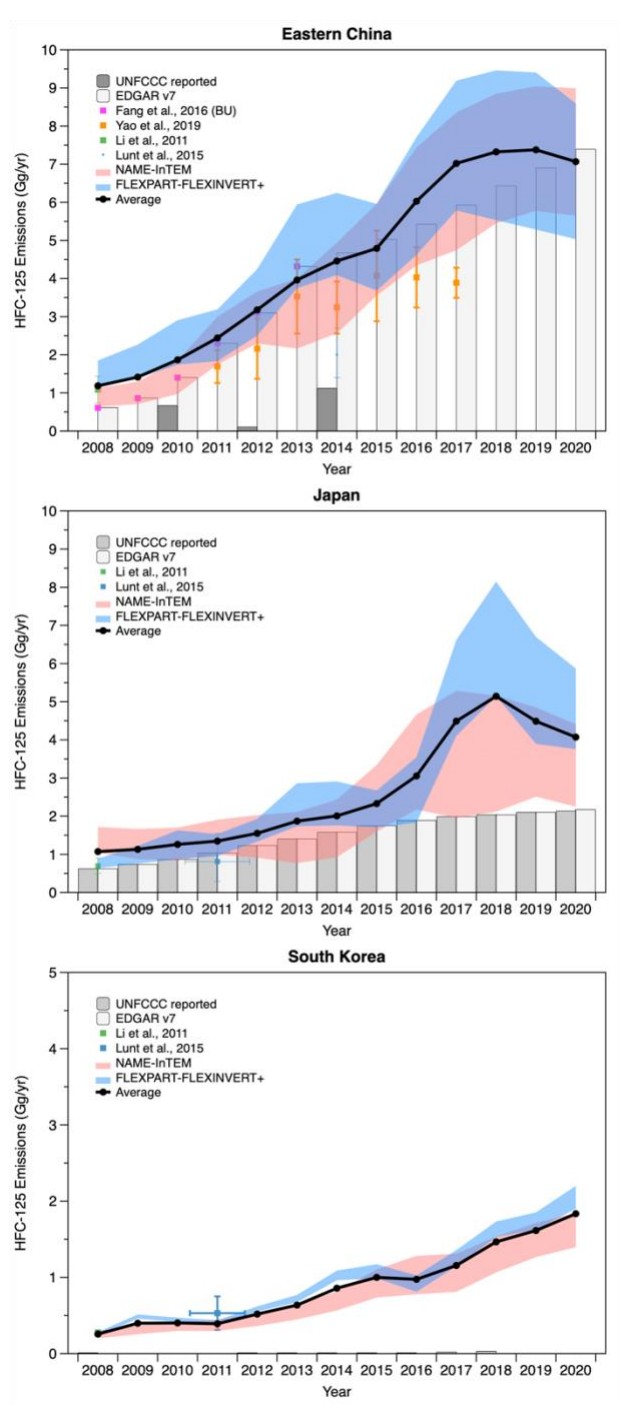

**Figure 6: Same as Fig. 4, but for HFC-125.**





### 3.2.4 HFC-143a

HFC-143a is used in refrigeration blends such as R-404A (44% of HFC-125, 52% of HFC-143a, and 4% of HFC-134a), an alternative refrigerant to the ozone depleting R-502 (48.8% of HCFC-22 and 51.2% of CFC-115). R-404A is mainly used in low and medium temperature refrigeration such as commercial refrigeration, supermarket display cases, etc. Among the five HFCs discussed here, HFC-143a is the least emitted in eastern Asia, but emissions have continuously increased from 2008 to 2020 in most countries: China (0.3 to 1.3 Gg/yr), South Korea (0.1 to 0.4 Gg/yr), North Korea (0.02 to 0.08 Gg/yr), and Taiwan (0.05 to 0.18 Gg/yr). HFC-143a emissions in Japan declined gradually from 0.5 to 0.33 Gg/yr between 2008 and 2012, but rapidly increased, peaking at 1.3 Gg/yr by 2018, followed by a stabilization. In contrast, Japan reported very low emissions increasing from 2008 to 2020 (0.02 to 0.06 Gg/yr) to the UNFCCC, showing a significant discrepancy (Figure 7). Similarly, Chinese emissions reported to UNFCCC downscaled to Eastern China, were much lower than the top-down emissions inferred here.



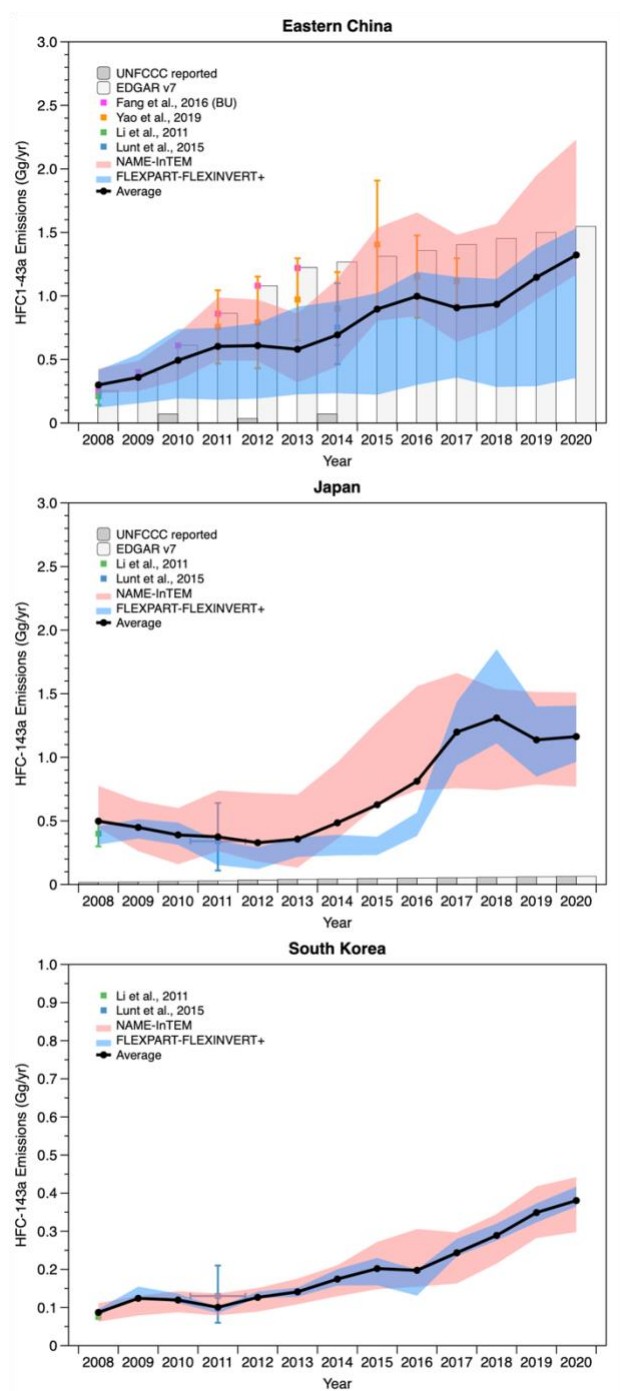

**Figure 7: Same as Fig. 4, but for HFC-143a.**



### 3.2.5 HFC-152a

HFC-152a, which has the lowest GWP (148) among HFCs, is primarily used as an aerosol propellant and insulation foam blowing agent (Montzka and Velders et al., 2018; Liang and Rigby et al., 2022). In eastern China, HFC-152a emissions were relatively constant from 2008 to 2014, on average $2.0 \pm 0.2$ Gg/yr with slight fluctuations, and then gradually increased from 2015 towards to 3.6 Gg/yr in 2019, before declining in 2020. Although eastern China shows the highest HFC-152a emissions in eastern Asia, the rate of increase in its emissions was relatively low compared to that of other HFCs. However, similar to

other HFCs, Chinese emissions reported to UNFCCC downscaled to eastern China, were much lower, almost close to zero, again revealing large discrepancies (Figure 8). On the other hand, EDGAR estimated much higher emissions during that period (almost twice the top-down emissions). HFC-152a emissions in Japan were almost constant from 2008–2014 at $0.8 \pm 0.1$ Gg/yr, then increased very steeply after 2015, reaching 3.4 Gg/yr in 2019, before declining in 2020. After the rapid increase Japan emissions since 2015, emissions of Japan and eastern China became nearly identical during 2017–2020. Japanese bottom-up

emissions reported to the UNFCCC were declining in contrast to our top-down inferred emissions. The gap (top-down minus bottom-up) has widened from -0.6 Gg/yr in 2008 to 3.2 Gg/yr in 2019. HFC-152a emissions in South Korea from 2008 to 2012 were negligible, with an average of $0.06 \pm 0.01$ Gg/yr. However, a significant increase in emissions was found from 2013 to 2018, reaching 1.4 Gg/yr in 2018, followed by a slower increase. As a result, South Korea showed the most substantial emissions growth of HFC-152a among all countries. During the period 2008-2013, when South Korea's inferred emissions

were near zero, there was a good match with emissions reported to UNFCCC, but after 2014, the inferred emissions are only about 50% of the emissions reported to the UNFCCC, that is UNFCCC reported emissions seem to be too high. North Korea had almost no HFC-152a emissions until 2015, followed by a slight increase and a plateau since 2017, averaging $0.4 \pm 0.1$ Gg/yr. Taiwan emitted negligible amounts of HFC-152a.





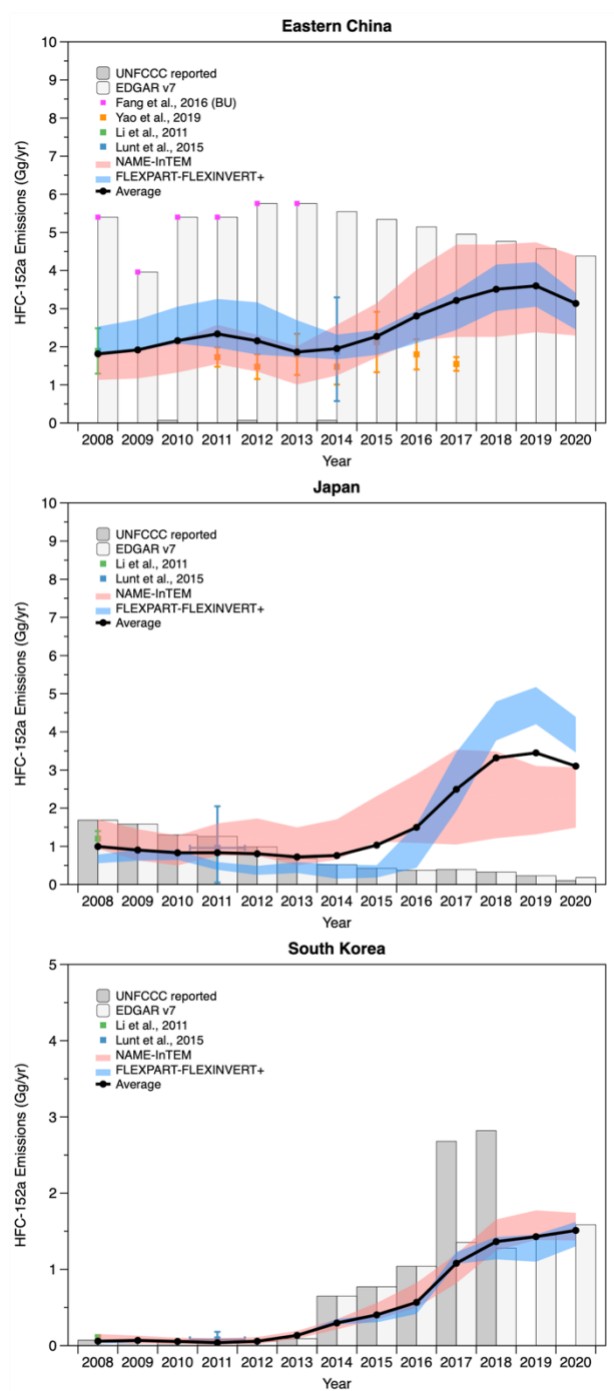

**Figure 8: Same as Fig. 4, but for HFC-152a.**



## 3.3 Spatial distributions of HFCs emissions in eastern Asia

As described in Section 3.1, estimating national-scale emissions via atmospheric inversion plays an important role in discovering and recognizing gaps in the bottom-up inventory, which is determined by sectoral reports, statistics, emission
factors, and other activity data (Lunt et al., 2015). Ideally this will help to improve bottom-up inventories and inventory methods. Even more insights into sources can be gained by examining the spatial distribution of gridded emissions from Bayesian inversion frameworks (Kunik et al., 2019).

Figure 9 depicts the spatial distribution of HFC emissions in eastern Asia, generally showing significant increases from 2008–2014 to 2016–2020.

The spatial distribution of HFC emissions showed significant emission increases likely stemming from increased use. For instance, HFC-134a, primarily used as a refrigerant for MAC systems, exhibits higher emissions in densely populated, large cities (e.g., Beijing, Shanghai, Nanjing, etc.). On the other hand, HFC-32 and HFC-125, which are used as components of refrigerant blends for air conditioning systems, show higher emissions in regions and provinces such as Shangdong and Anhui with high levels of economic, commercial, and industrial activities. Likewise, HFC-143a, utilized for low and medium
temperature ranges such as refrigerated transport and supermarket freezers/refrigerators, displays higher emissions in areas with considerable economic, commercial, and logistics transportation activities like Shanghai. Lastly, HFC-152a, predominantly used as a foam blowing agent and aerosol propellant, exhibits higher emissions in regions near province of Shangdong with significant industrial activities. In addition, although not included in the HFCs emissions for eastern China presented in this study, distinct post-2015 increases in emissions of all HFCs, except HFC-143a, were also revealed in the
Changchun and Harbin regions of northeast China. Furthermore, China is the largest producer and exporter of HFCs in the world (Fang et al., 2016), and given the presence of many production plants in eastern China with the capacity to produce various HFC species simultaneously, it is necessary to consider the possibility of an increase in fugitive emissions resulting from the growing HFCs production in eastern China (Li et al., 2014; Velders et al., 2022). South Korea (Greater Seoul and Busan) and Japan (Kansai, Kanto, and Subu regions) showed a noticeable increase in all HFCs emissions as well.



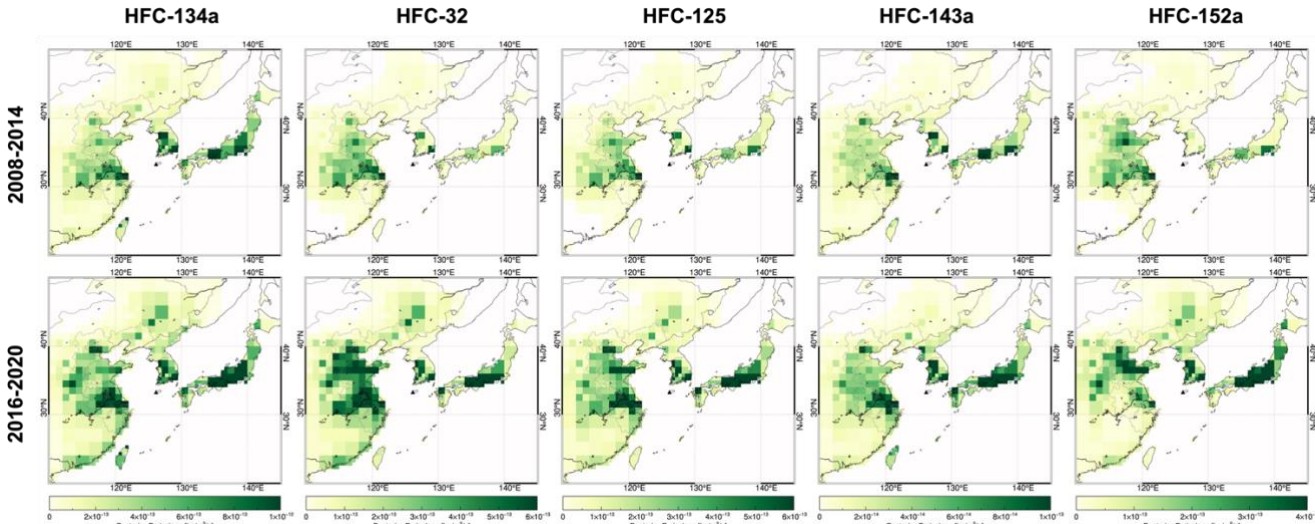


**Figure 9: HFCs emissions distributions for 2008–2014 and 2016–2020 derived by FLEXPART-FLEXINVERT+. The results of NAME-InTEM are shown in Fig. S7.**

## 3.4 Aggregated HFCs emissions in eastern Asia and proportional contributions by each country.

Total national emissions of the five major HFCs in eastern Asia in terms of carbon dioxide equivalent emissions (using $GWP_{100}$) increased from 20.7 Gg/yr (32.2 $CO_2$-eq Tg/yr) in 2008 to a peak of 71.3 Gg/yr (116.2 $CO_2$-eq Tg/yr) in 2018, before declining to 65.6 Gg/yr (109.8 $CO_2$-eq Tg/yr) in 2020 (Figure 10). The increase in total HFC emissions in eastern Asia during 2008-2018 was driven by Japan and eastern China in particular, with eastern China emitting about 1.1 Gg/yr (4.1 $CO_2$-eq Tg/yr) of total HFCs less than Japan in 2008. However, since 2010, the emissions of eastern China have exceeded those of Japan. From

2019 onwards, total HFC emissions in eastern Asia started to decline as a result of the stagnation in emissions growth in eastern China since 2016 and the gradual decline of total HFC emissions in Japan, despite the continued growth in emissions in South Korea.

Figure 10 shows that in terms of $CO_2$-equivalent emissions ($GWP_{100}$), HFC-125 and HFC-134a contribute the most to the total emissions in eastern Asia. Although HFC-143a is emitted in the smallest quantity, due to its highest GWP of 5900 among the

five HFCs, it showed a larger warming impact compared to HFC-32, which is emitted in higher quantities, but has a lower GWP of 749. On the other hand, despite its increasing emissions, the impact of HFC-152a on global warming is negligible due to the relatively low GWP of 148.



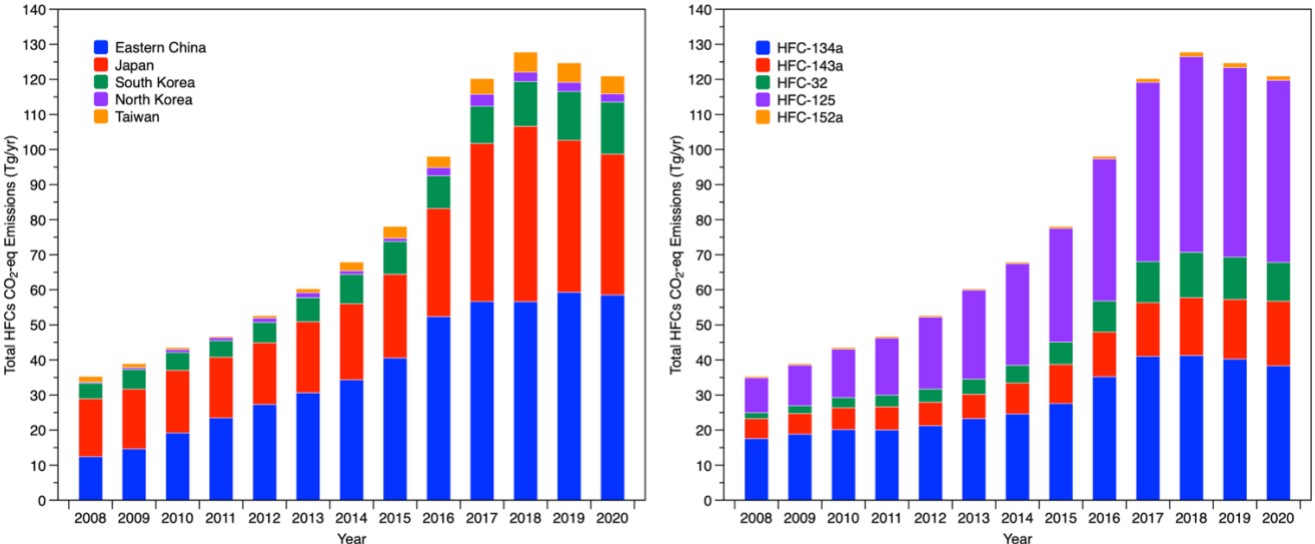


**Figure 10: Total HFCs CO₂-eq emissions (in Tg/yr) in eastern Asia for each region and each HFC, respectively.**

In addition to identifying the total HFC emissions of each country, the variation in the proportion of each HFC in total HFC emissions by country can be important information for establishing a regulation policy to mitigate GHGs. For example, if a

country has a significant proportion of a specific HFC, it can proactively identify the underlying causes, implement measures to restrict its usage, enhance recycling efficiency, or explore alternative options.

Especially, as an Annex-I country, Japan has a more proactive plan to reduce HFCs emissions than other countries in eastern Asia. In April 2015, the Japanese government implemented a revised F-Gas Law, introducing new policies to restrict the utilization of fluorinated refrigerants throughout the entire life cycle, including manufacturing, usage, and disposal of

fluorocarbons and products containing them. This expansion in scope shifted the focus from solely recovering and destroying F-gases to encompassing various stages, such as manufacturing, maintenance, leak checking, and the promotion of low-GWP or non-fluorocarbon refrigerants in designated products. Target GWPs and target years have been determined for each product category, and the transition to low-GWPs is underway, starting with Residential Air Conditioners in 2018 and continuing through the mid-2020s for all categories. Therefore, we calculated the change in the proportion of each HFC's emissions in

each region's total HFC CO₂-equivalent emissions for 2008, 2014, and 2020 (Figure 11). The HFC emission results in mass perspective are presented in Figure S8.

CO₂-equivalent HFCs emissions show a different proportion than that of mass (see Appendix/Supplement). In 2008, more than half of the total HFC emissions in eastern Asia were accounted for by HFC-134a, which had the largest portion in each country. However, since then the emissions of HFC-125, which has the highest GWP among the five HFCs, have increased rapidly,

and in 2020, HFC-125 has become the most dominant HFC in terms of its contribution to global warming. These changes have been more pronounced in South Korea and Japan, where the proportion of HFC-125 has increased by approximately two-fold in South Korea from 22% in 2008 to 47% in 2020, while HFC-134a has decreased by almost half from 63% to 28%. Similarly,



in Japan, HFC-125 increased from 25% to 39%, while HFC-134a decreased from 54% to 36%. In contrast, in eastern China, the changes in the percentage of each HFC over the years have been relatively small. Since Japan's F-gas Law was revised and

the transition to low GWP began in 2018, it is crucial to monitor the changes in the proportion of low-GWP HFCs as well as the total amount of HFC emissions. The changes in the proportion of HFCs can be reflected by each country's industrial structure and policies, and based on this, it may be effective for each country to pursue policies to develop and transition to environmentally friendly alternatives, focusing on HFC-125 and HFC-134a, to respond to global warming.

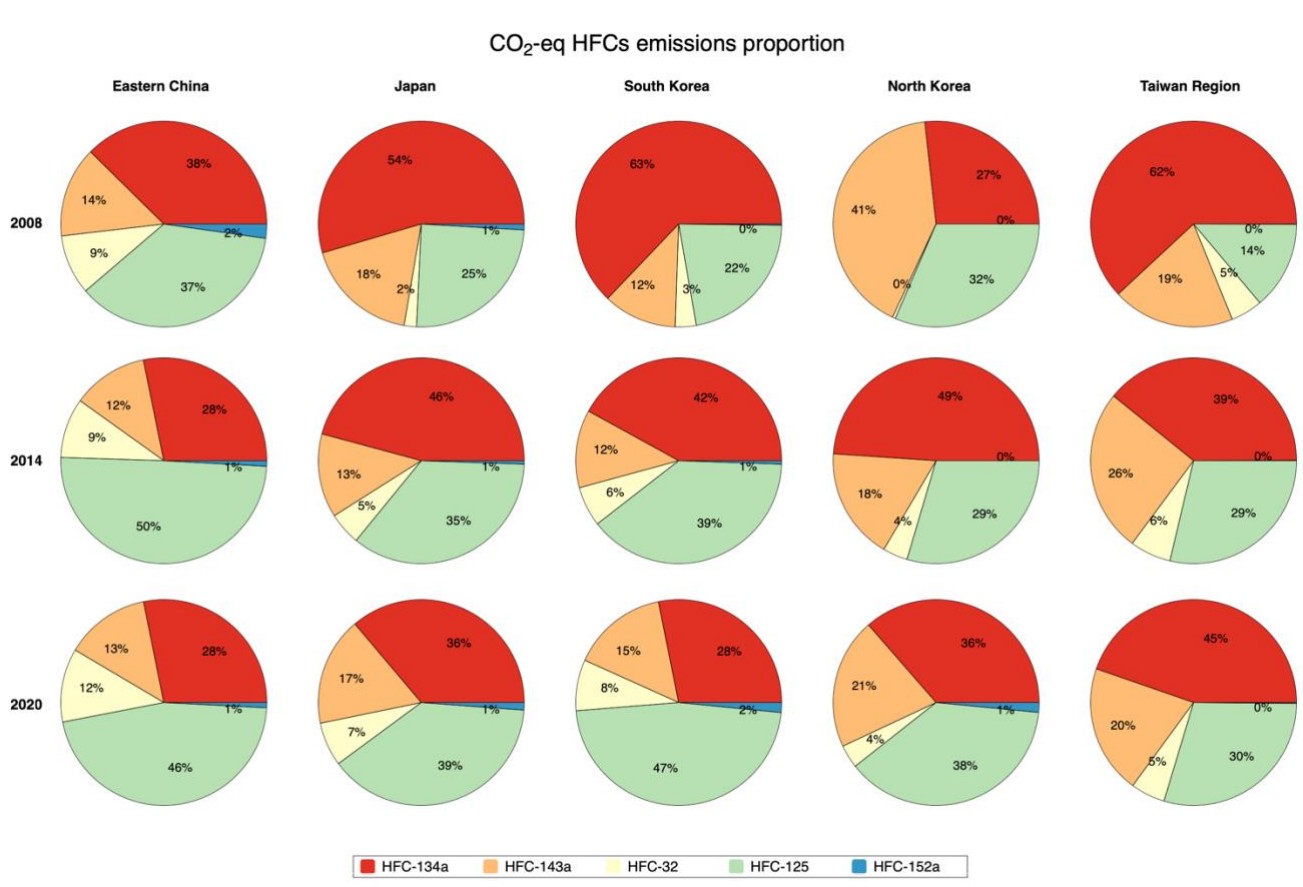


**Figure 11: The proportion of each HFC CO₂-equivalent emission in each country for 2008, 2014, and 2020.**

**3.5 Emissions contribution of eastern Asia in a global perspective**

As shown in Figure 12, global total HFCs emissions (Liang and Rigby et al., 2022) that were derived based on the observations from the AGAGE network using AGAGE 12-box model (Rigby et al., 2013) have steadily increased from 249.0 ± 21.6 Gg/yr



(436.8 ± 30.8 $CO_2$-eq Tg/yr) in 2008 to 495.2 ± 50.9 Gg/yr (961.2 ± 89.4 $CO_2$-eq Tg/yr) in 2020. On the other hand, aggregated HFCs emissions reported to the UNFCCC by Annex-I countries have increased steadily, with very small fluctuation, from

128.5 Gg/yr (256.0 $CO_2$-eq Tg/yr) in 2008 to 159.4 Gg/yr (342.1 $CO_2$-eq Tg/yr) in 2020. (Average total HFC emissions from 2008–2020 are ~149.6 ± 10.0 Gg/yr (286.7 ± 26.4 $CO_2$-eq Tg/yr)). As a result, the unreported emissions, which correspond to gaps between the global top-down and reported bottom-up (UNFCCC) emissions from Annex-I countries, increased from 120.5 Gg/yr (180.8 ± 30.8 $CO_2$-eq Tg/yr) in 2008 to 335.9 Gg/yr (561.7 ± 82.1 $CO_2$-eq Tg/yr) in 2020, indicating that the increasing trend in unreported emissions, as noted in previous studies (Rigby et al., 2014 and Velders et al., 2022), continues

through to 2020.

The contribution of eastern Asia's total HFCs $CO_2$-eq emissions to global emissions was relatively constant for 2008–2014 with a mean of 8.9 ± 0.7%, and then increased rapidly to a peak of 14.4% in 2018, then slightly declined to 12.6% in 2020. This indicates that the growth rate of HFCs emissions in eastern Asia has been faster than that globally since 2015. Of these, HFC emissions from non-Annex-I countries in eastern Asia (i.e., eastern China, South Korea, North Korea, Taiwan), have

significantly contributed to global unreported emissions. The proportion has steadily increased from 10.5% in 2008, peaked at 14.7% in 2016, and then slightly decreased to 13.1% in 2020, and it accounts for 13.3% of the unreported accumulated HFCs emissions during the period of 2008–2020. It is clear that substantial emissions from other regions, not reporting to UNFCCC and/or not being covered by atmospheric measurements must exist.




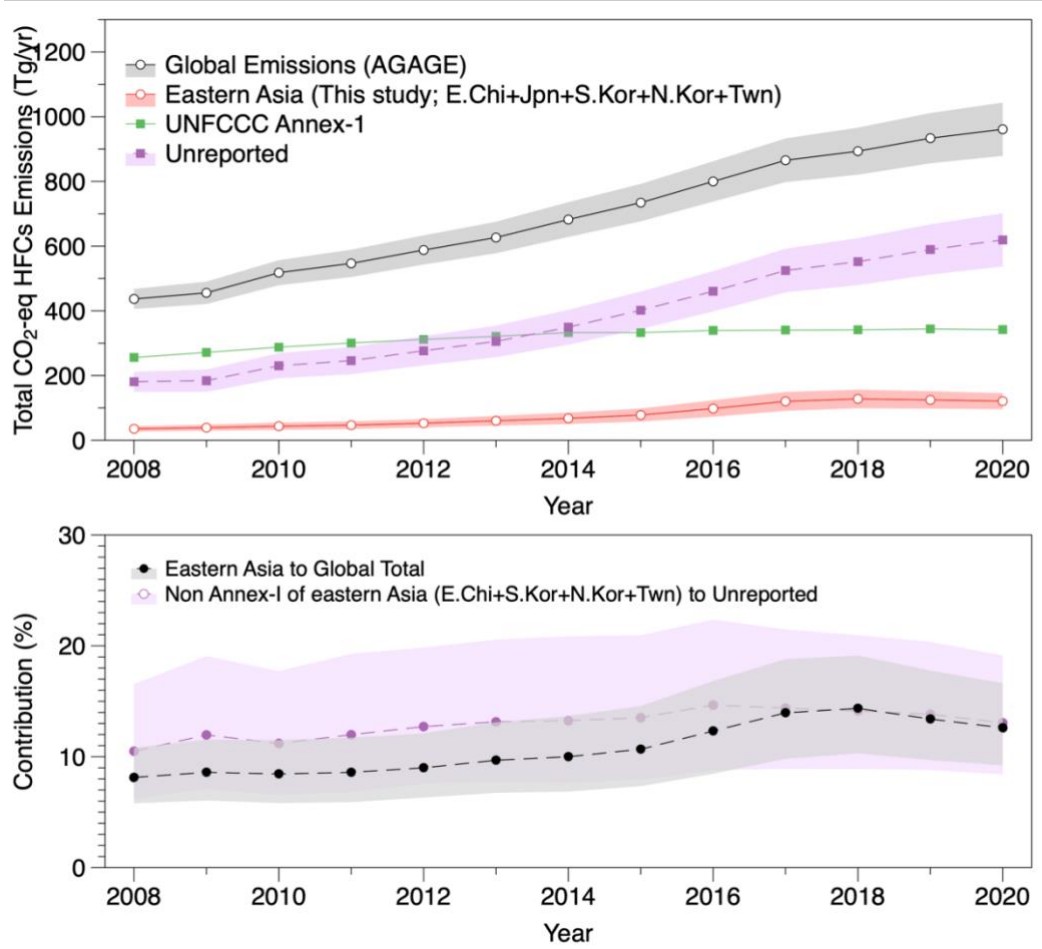

**Figure 12: (top) Annual global total emissions of HFCs derived from atmospheric observations of the AGAGE program from 2008 to 2020 (black), inferred regional emissions in eastern Asia (red), (bottom) Proportional contributions of HFCs emissions in eastern Asia to the global total (Results for individual HFCs are shown in Figures S9 to S14).**



## 4 Summary and Conclusions

HFCs are potent, anthropogenic, greenhouse gases that contribute to global warming and continue to be emitted due to industrial growth, however, a phase-down of consumption and production started under the Kigali amendment. We analyzed high precision atmospheric mole fractions of major HFCs (HFC-134a, HFC-32, HFC-125, HFC-143a and HFC-152a) measured at the Gosan station from 2008 to 2020 to understand the temporal trends of HFCs emissions in eastern Asia. The background mole fractions of most HFCs continue to increase (with a slight increase for HFC-152a) and it suggests that there are ongoing emissions from surrounding sources within this region (of note, the enhancement magnitudes of most HFCs show rapid growth). From the observations, we inferred annual HFCs emissions estimates for eastern China, western Japan, South Korea, North Korea, and Taiwan by using the two independent Bayesian inversion frameworks, FLEXPART-FLEXINVERT+ and UK Met Office's NAME-InTEM, and compared to bottom-up inventories reported to UNFCCC and EDGAR-v7 (for the comparable countries and substances). All HFCs emissions have shown a notable rapid increase in eastern Asia.

Our most notable findings are:

i. Rapid increase in HFC emissions in eastern Asia

HFCs emissions in eastern Asia were 2–3 times (HFC-134a, HFC143a, and HFC-152a) and 5–6 times (HFC-32 and HFC-125) higher in 2020 than in 2008, with emissions growth driven predominantly in eastern China, Japan, and South Korea. Emissions of HFCs have an accelerating upward trend since 2015. Based on the average spatial distribution of HFCs emissions between 2008 and 2014 compared to 2016 and 2020, the increased emissions are associated with HFCs production facilities as well as the usage of each HFC compound over megacities and industrially and commercially active regions. The annual quantification of HFCs emissions indicates that HFC-134a and HFC-143a emissions from eastern China and Japan are similar. However, for HFC-32 and HFC-125, eastern China's emissions are twice as those of Japan. For HFC-152a, Japan's emissions amounted to half of eastern China's emissions up to 2015, but following a rapid increase in emissions from Japan after 2016, both countries have similar levels of emissions from 2018 onwards. In the case of South Korea, all HFCs emissions have shown a steadily annual increase. Although quantitative emissions of South Korea are small compared to eastern China and Japan, per capita emissions in South Korea are approximately 0.3–0.5 times that of Japan and 0.7–2.3 times that of China, which is not negligible. Before the acceleration started in 2015, total HFC emissions from eastern Asia accounted for ~9% of global emissions, while post-2015, this proportion increased to ~15%. Additionally, these total cumulative HFCs emissions of non-Annex-I countries in eastern Asia from 2008 to 2020 accounted for approximately 13% of the unreported emissions, highlighting the significance of eastern Asia as a region from a global perspective. Throughout the entire period, HFC-134a has been the most emitted compound in eastern Asia among the 5 major HFCs. However, considering its contribution to $CO_2$-eq emissions from a global warming perspective, the emissions of HFC-125, which has a higher GWP, showed a rapid increase and surpassed the global warming impact of HFC-134a emissions.





ii. Unexpected increase in HFC emissions in Japan

Japan, an Annex-I country, has taken a more advanced approach to reducing HFC emissions than other countries in eastern Asia by revising Japan's F-Gas law. However, contrary to the expectation that recent emissions of HFCs in Japan would be maintained or decreased, the current trends in HFC emissions in Japan, as derived from this study, remained relatively constant (or increased very slowly) before 2015 and had sharply increased after 2015 until the peak in 2018. The reason for this unexpected increase in emissions of HFCs is not clear but may be related to the recent increase in sales of refrigerants in the

domestic market of Japan, as compiled by the Japan Refrigeration and Air conditioning Industry Association (JRAIA). As shown in Figure S14, from 2015 to 2019, compared to 2008, there was a notable increase in domestic demand for commercial air conditioners (31%), refrigerators (30%), spot air conditioners (130%), and residential air conditioners (27%), as well as an increase in demand for transportation refrigeration and freezer units (36%) and bus air conditioners (7%), respectively. The high consumption rates for the mid-to-late 2010s in the various refrigerant-usage sectors may imply that large quantities of

outdated equipment had been disposed of and replaced, likely resulting in an unexpected increase in HFC emissions in Japan. Nevertheless, the downward trend in Japan's HFCs emissions since 2019 may also reflect the effectiveness of Japan's F-gas policy, which restricts the use of fluorinated refrigerants and products made with them at each stage of manufacture and use.

iii. Discrepancies between the observation-derived top-down emissions and the reported bottom-up inventories

As an Annex-1 country, Japan has an obligation to report its annual HFCs emissions to UNFCCC. The reported inventories of HFC-134a and HFC-143a by Japan to the UNFCCC were slightly lower than the observation-derived top-down estimates. In contrast, HFC-32, HFC-125, and HFC-152a presented relatively similar emissions within the uncertainty range with the reported inventories up to 2015. However, after 2015, there increasing differences. Furthermore, while HFC-152a estimates reported to the UNFCCC showed a decrease in emissions over time, the top-down emissions estimates exhibited a sharp

increase. Despite not being obligated to report, China and South Korea have voluntarily reported national emissions of specific substances to the UNFCCC for a limited number of years. In contrast to Japan, the top-down emissions for China and South Korea for HFC-134a were approximately half of reported bottom-up inventories, and for other HFCs (except for HFC-152a of South Korea), reported emissions were either very low or not available due to lack of the statistical information of production or consumption. In particular, the reason for the very low reported (or not reported) emissions of HFC-32, HFC-125, and HFC-

143a in South Korea is that the national inventory compilation does not yet consider the amount of HFCs contained in blended refrigerants such as R-410A and R-404A (The Government of the Republic of Korea, the fourth biennial update report of the Republic of Korea; https://unfccc.int/documents/418616, last access: 28 Nov 2023). Therefore, the consideration of blended refrigerant components is also necessary when estimating the bottom-up emissions of each HFCs. These indicate the importance of comprehensive data collection and accurate reporting to ensure precise estimation and management of emissions,

both in developed countries where some emissions may be overlooked, as well as in developing countries where the establishment of robust bottom-up emission estimation requires accurate information on production and consumption.



Furthermore, as it was revealed the significant discrepancies between top-down and bottom-up approaches in Japan for specific substances or periods, it becomes necessary to compare and evaluate the reported emissions from other developed countries with the emissions that have been reported or will be reported.


Despite the continuing upward trend in HFC emissions in eastern Asia, the decline in Japan since 2019 likely reflects the effectiveness of Japan's F-gas policy, while the decline in eastern China in 2020 can perhaps be attributed to the lockdown measures implemented in response to the COVID-19 pandemic. As such, the variation of HFCs emissions in eastern Asia is increasingly significant for global hydrofluorocarbon emissions and their impact on mitigating future climate change. Indeed,
it is crucial to address the growing demand for HFCs in eastern Asia countries in a way that balances economic growth with the need for a gradual reduction in HFCs emissions. To this end, policies are being formulated and implemented to promote the transition to low-GWP alternatives and improve energy efficiency in the industrial sector in various countries. Overall, the regulation and phase-down of HFC emissions is an important effort to comply with the Kigali Amendment and combat climate change. Therefore, we will continue to monitor atmospheric abundance of HFCs and estimate quantitative emissions at the
regional scale to understand the changing trends of HFCs emissions in each country in eastern Asia.

**Data availability**

Data used in this study are available from the AGAGE (Advanced Global Atmospheric Gases Experiment) database (http://agage.eas.gatech.edu/data_archive/agage/gc-ms-medusa/, last access: May 2023). NAME and InTEM are available for research use from the UK Met Office and subject to licence.

**Author contributions**

HC, and SP designed the study; HC, ALR, JM, RT, and AJM interpreted the analyzed results; HC, ALR, RT, JM, SP, and JK wrote and revised the manuscript; HC, SP, and HP carried out the measurement of HFCs at Gosan; JM, PKS, CMH and RFW supported the calibration and long-term precision for the observations at Gosan.

**Competing interests**

The authors declare that they have no conflict of interest.

**Acknowledgements**

This research was supported by the National Research Foundation of Korea (NRF) grant funded by the Korean government (MSIT) (no. 2021R1I1A1A01045062). Support for contributions by J. Kim, J. Mühle, C. M. Harth, P. K. Salameh, and R. F. Weiss came from National Aeronautics and Space Administration (grant nos. NNX16AC96G and NNX16AC97G). A. L.



Redington and A. J. Manning were supported by the Met Office Hadley Centre Climate Programme funded by UK government departments BEIS and Defra.

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



Table 3: Annual averages of HFCs emissions estimates derived from FLEXPART-KNU and NAME-InTEM.

| Compounds / Regions [Gg/yr] | Year | | | | | | | | | | | | |
|---|---|---|---|---|---|---|---|---|---|---|---|---|---|
| | 2008 | 2009 | 2010 | 2011 | 2012 | 2013 | 2014 | 2015 | 2016 | 2017 | 2018 | 2019 | 2020 |
| **HFC-134a** | | | | | | | | | | | | | |
| E. China | 3.18 ± 0.77 | 3.65 ± 0.87 | 4.68 ± 1.23 | 5.35 ± 1.52 | 5.84 ± 1.44 | 6.06 ± 1.54 | 6.58 ± 1.77 | 8.52 ± 2.07 | 11.71 ± 2.91 | 11.59 ± 2.94 | 10.30 ± 2.57 | 11.11 ± 2.70 | 11.21 ± 2.55 |
| Japan | 6.12 ± 1.27 | 6.45 ± 1.45 | 6.76 ± 1.65 | 6.20 ± 1.47 | 5.98 ± 1.40 | 6.75 ± 1.71 | 6.76 ± 1.70 | 6.59 ± 1.72 | 8.16 ± 2.38 | 11.46 ± 2.73 | 12.35 ± 2.45 | 10.83 ± 2.18 | 9.89 ± 1.89 |
| S. Korea | 1.90 ± 0.19 | 2.11 ± 0.19 | 1.78 ± 0.19 | 1.62 ± 0.20 | 1.90 ± 0.21 | 2.16 ± 0.22 | 2.40 ± 0.22 | 2.49 ± 0.32 | 2.52 ± 0.47 | 2.62 ± 0.46 | 2.88 ± 0.34 | 2.95 ± 0.32 | 2.85 ± 0.36 |
| N. Korea | 0.06 ± 0.22 | 0.11 ± 0.22 | 0.28 ± 0.23 | 0.36 ± 0.26 | 0.47 ± 0.31 | 0.49 ± 0.43 | 0.33 ± 0.41 | 0.20 ± 0.41 | 0.59 ± 0.64 | 0.97 ± 0.78 | 0.75 ± 0.72 | 0.64 ± 0.56 | 0.57 ± 0.53 |
| Taiwan | 0.70 ± 0.34 | 0.48 ± 0.35 | 0.19 ± 0.36 | 0.10 ± 0.42 | 0.25 ± 0.43 | 0.36 ± 0.45 | 0.67 ± 0.83 | 0.95 ± 1.03 | 0.97 ± 0.85 | 1.27 ± 0.75 | 1.79 ± 0.71 | 1.86 ± 0.65 | 1.55 ± 0.60 |
| **HFC-32** | | | | | | | | | | | | | |
| E. China | 1.6 ± 0.34 | 1.95 ± 0.36 | 2.59 ± 0.50 | 3.13 ± 0.67 | 3.55 ± 0.69 | 3.85 ± 0.74 | 4.31 ± 0.89 | 5.47 ± 1.05 | 7.72 ± 1.29 | 9.28 ± 1.43 | 9.93 ± 1.90 | 9.95 ± 2.08 | 9.03 ± 1.76 |
| Japan | 0.43 ± 0.27 | 0.61 ± 0.27 | 0.87 ± 0.26 | 0.96 ± 0.26 | 1.06 ± 0.29 | 1.29 ± 0.41 | 1.48 ± 0.50 | 1.93 ± 0.63 | 2.89 ± 0.97 | 4.84 ± 1.66 | 5.32 ± 1.87 | 4.10 ± 1.50 | 3.70 ± 1.23 |
| S. Korea | 0.20 ± 0.03 | 0.31 ± 0.03 | 0.33 ± 0.03 | 0.32 ± 0.03 | 0.40 ± 0.04 | 0.53 ± 0.05 | 0.71 ± 0.07 | 0.82 ± 0.11 | 0.85 ± 0.15 | 1.02 ± 0.16 | 1.35 ± 0.16 | 1.51 ± 0.17 | 1.59 ± 0.18 |
| N. Korea | 0.00 ± 0.03 | 0.01 ± 0.03 | 0.01 ± 0.04 | 0.01 ± 0.05 | 0.02 ± 0.06 | 0.05 ± 0.07 | 0.04 ± 0.08 | 0.04 ± 0.14 | 0.09 ± 0.22 | 0.11 ± 0.26 | 0.17 ± 0.24 | 0.21 ± 0.22 | 0.11 ± 0.21 |
| Taiwan | 0.11 ± 0.19 | 0.08 ± 0.12 | 0.04 ± 0.06 | 0.03 ± 0.08 | 0.03 ± 0.11 | 0.09 ± 0.13 | 0.22 ± 0.23 | 0.28 ± 1.28 | 0.30 ± 0.26 | 0.46 ± 0.27 | 0.47 ± 0.31 | 0.38 ± 0.30 | 0.37 ± 0.26 |
| **HFC-125** | | | | | | | | | | | | | |
| E. China | 1.19 ± 0.30 | 1.41 ± 0.37 | 1.87 ± 0.51 | 2.44 ± 0.65 | 3.18 ± 0.78 | 3.96 ± 1.01 | 4.46 ± 1.13 | 4.79 ± 1.17 | 6.03 ± 1.55 | 7.02 ± 1.75 | 7.32 ± 1.83 | 7.38 ± 1.85 | 7.07 ± 1.73 |
| Japan | 1.07 ± 0.23 | 1.13 ± 0.31 | 1.26 ± 0.40 | 1.35 ± 0.37 | 1.55 ± 0.43 | 1.87 ± 0.62 | 2.01 ± 0.67 | 2.33 ± 0.68 | 3.05 ± 1.05 | 4.49 ± 1.46 | 5.14 ± 1.51 | 4.49 ± 1.29 | 4.07 ± 1.07 |
| S. Korea | 0.26 ± 0.04 | 0.40 ± 0.04 | 0.40 ± 0.04 | 0.39 ± 0.05 | 0.52 ± 0.06 | 0.64 ± 0.07 | 0.86 ± 0.09 | 1.00 ± 0.14 | 0.97 ± 0.18 | 1.16 ± 0.18 | 1.47 ± 0.17 | 1.61 ± 0.17 | 1.84 ± 0.19 |
| N. Korea | 0.03 ± 0.05 | 0.04 ± 0.05 | 0.05 ± 0.06 | 0.05 ± 0.07 | 0.07 ± 0.10 | 0.09 ± 0.11 | 0.08 ± 0.13 | 0.11 ± 0.19 | 0.28 ± 0.28 | 0.33 ± 0.35 | 0.24 ± 0.32 | 0.26 ± 0.27 | 0.23 ± 0.24 |
| Taiwan | 0.06 ± 0.08 | 0.06 ± 0.08 | 0.03 ± 0.08 | 0.03 ± 0.11 | 0.03 ± 0.15 | 0.08 ± 0.17 | 0.19 ± 0.27 | 0.26 ± 0.33 | 0.27 ± 0.29 | 0.38 ± 0.29 | 0.43 ± 0.29 | 0.40 ± 0.28 | 0.39 ± 0.26 |





| | Year | | | | | | | | | | | | |
| --- | --- | --- | --- | --- | --- | --- | --- | --- | --- | --- | --- | --- | --- |
| | 2008 | 2009 | 2010 | 2011 | 2012 | 2013 | 2014 | 2015 | 2016 | 2017 | 2018 | 2019 | 2020 |
| **HFC-143a** | | | | | | | | | | | | | |
| E. China | 0.30 ± 0.12 | 0.36 ± 0.16 | 0.49 ± 0.23 | 0.60 ± 0.27 | 0.61 ± 0.27 | 0.58 ± 0.31 | 0.69 ± 0.35 | 0.90 ± 0.38 | 1.00 ± 0.43 | 0.91 ± 0.41 | 0.93 ± 0.42 | 1.15 ± 0.52 | 1.32 ± 0.56 |
| Japan | 0.50 ± 0.12 | 0.45 ± 0.14 | 0.39 ± 0.15 | 0.37 ± 0.17 | 0.33 ± 0.18 | 0.36 ± 0.18 | 0.48 ± 0.19 | 0.63 ± 0.20 | 0.81 ± 0.25 | 1.20 ± 0.35 | 1.31 ± 0.38 | 1.14 ± 0.32 | 1.16 ± 0.30 |
| S. Korea | 0.09 ± 0.02 | 0.12 ± 0.02 | 0.12 ± 0.02 | 0.10 ± 0.02 | 0.13 ± 0.02 | 0.14 ± 0.02 | 0.17 ± 0.03 | 0.20 ± 0.05 | 0.20 ± 0.06 | 0.24 ± 0.05 | 0.29 ± 0.04 | 0.35 ± 0.05 | 0.38 ± 0.05 |
| N. Korea | 0.02 ± 0.03 | 0.03 ± 0.03 | 0.04 ± 0.03 | 0.02 ± 0.03 | 0.04 ± 0.04 | 0.05 ± 0.05 | 0.03 ± 0.05 | 0.03 ± 0.07 | 0.05 ± 0.09 | 0.10 ± 0.10 | 0.08 ± 0.09 | 0.08 ± 0.08 | 0.08 ± 0.09 |
| Taiwan | 0.05 ± 0.04 | 0.04 ± 0.05 | 0.02 ± 0.05 | 0.02 ± 0.05 | 0.03 ± 0.06 | 0.05 ± 0.07 | 0.11 ± 0.13 | 0.14 ± 0.15 | 0.10 ± 0.10 | 0.15 ± 0.19 | 0.19 ± 0.10 | 0.17 ± 0.10 | 0.18 ± 0.10 |
| **HFC-152a** | | | | | | | | | | | | | |
| E. China | 1.81 ± 0.37 | 1.92 ± 0.37 | 2.16 ± 0.45 | 2.34 ± 0.58 | 2.16 ± 0.58 | 1.86 ± 1.01 | 1.95 ± 0.49 | 2.27 ± 0.52 | 2.81 ± 0.68 | 3.21 ± 0.86 | 3.51 ± 0.91 | 3.60 ± 0.88 | 3.14 ± 0.76 |
| Japan | 1.00 ± 0.25 | 0.90 ± 0.31 | 0.83 ± 0.26 | 0.83 ± 0.26 | 0.80 ± 0.30 | 0.72 ± 0.62 | 0.76 ± 0.36 | 1.03 ± 0.38 | 1.50 ± 0.74 | 2.49 ± 1.00 | 3.32 ± 0.82 | 3.45 ± 0.69 | 3.10 ± 0.63 |
| S. Korea | 0.06 ± 0.03 | 0.07 ± 0.04 | 0.05 ± 0.02 | 0.04 ± 0.02 | 0.06 ± 0.03 | 0.13 ± 0.07 | 0.30 ± 0.05 | 0.40 ± 0.08 | 0.57 ± 0.11 | 1.08 ± 0.14 | 1.36 ± 0.18 | 1.43 ± 0.19 | 1.51 ± 0.17 |
| N. Korea | 0.00 ± 0.06 | 0.00 ± 0.05 | 0.00 ± 0.06 | 0.00 ± 0.06 | 0.00 ± 0.06 | 0.00 ± 0.11 | 0.00 ± 0.07 | 0.03 ± 0.11 | 0.14 ± 0.18 | 0.43 ± 0.30 | 0.40 ± 0.35 | 0.36 ± 0.26 | 0.23 ± 0.21 |
| Taiwan | 0.00 ± 0.10 | 0.00 ± 0.08 | 0.02 ± 0.08 | 0.02 ± 0.09 | 0.00 ± 0.11 | 0.00 ± 0.17 | 0.00 ± 0.23 | 0.00 ± 0.28 | 0.00 ± 0.20 | 0.00 ± 0.22 | 0.00 ± 0.24 | 0.01 ± 0.22 | 0.05 ± 0.17 |