# Peer review of "Revealing the significant acceleration of Hydrofluorocarbon (HFC) emissions in eastern Asia through long-term atmospheric observations"

_EGUsphere, 2023_

## Author Comment (AC1)

**Response to All reviewers**

We thank the referees for their thoughtful and thorough reviews. We are pleased that all the reviewers see our manuscript as a valuable contribution to the field. We have made changes to the manuscript to answer the suggestions of the reviewers and clarified a few points raised in review. A revised version of the manuscript including most of the changes suggested by the reviewers will be submitted to the editor. We thank the reviewers and the editor for their time and effort.

[In the following, Reviewers' comments are in black and our responses and are in red]

**Response to Reviewer #1**

We appreciate your very meaningful comments.

It gave us a deeper understanding of what we overlooked and didn't take into account, which enriched the manuscript.

SPECIFIC COMMENTS:

Title: The title focuses on the acceleration of HFC emissions, which was strongest from 2015 to 2018, but misses the subsequent decrease in emissions since 2019 which is also an important finding.

→ Yes, we fully agree with the reviewer's opinion. The decrease in emissions after 2019 is also an important point. One of notable founds in this study is the rapid increase in HFC emissions in East Asia. After 2019, while there has been a reduction in emissions in East Asia due to Japan's influence, the reduction is modest and the levels of HFC emissions are still higher than before the rapid increase. The decrease in Japan post-2019 requires ongoing monitoring and understanding. Therefore, instead of emphasizing post-2019 emissions in the title, we highlighted it in the subtitle of the conclusion section as "Unexpected increase in HFC emissions late 2010s and a recent decline in Japan".

Abstract:

One of the interesting results is the relative decrease in GWP contribution for HFC-134a and the relative increase for HFC-125 in most countries (Figure 11). Consider adding this to the abstract.

→ Revised. According to the opinion of reviewer, we added additional sentence at the end of the abstract. "Further, the proportional contribution of each country's $CO_2$-equivalent HFC emissions has changed over time, with HFC-134a decreasing and HFC-125 increasing. This demonstrates the transition in the predominant HFC substances contributing to global warming in each country".

L31: Check that 15% is correct. This seems to be the 2018 peak rather than the 2016-2020 average.

→ Thank you for pointing out the numerical error. Yes, it was the maximum value. The average for 2016-2020 is approximately 13%. The figure in the text has been corrected to 13%.

L33: Provide context for the 76 Gg/yr higher emissions. What are the total emissions?

→ Total emissions are 114 Gg/yr and emissions reported to UNFCCC are 38 Gg/yr. We specified those values in the manuscript.

L36: Reading about a decline in HFC emissions in 2019 is unexpected after reading about the sharp increase in emissions since 2016 (L30). It may help to change 'since 2016' to 'from 2016 to 2018'.

→ Revised. Since 2019 looked different, we changed 'since 2016' to 'from 2016 to 2018'.

L46: Provide a reference for the GWP of HFCs.

→ Revised. The reference Liang and Rigby et al. (2022) specified.

L54: State which of the 4 groups China, Japan, South Korea, North Korea and Taiwan are part of.

→ It is a good idea to mention a category group for each eastern Asia country. We think that classifying each eastern Asia country's status here could appear contextually awkward. Therefore, in the fifth paragraph, which mentioned to the Kigali Amendment and states, "As the reduction of HFC emissions is gradually being achieved globally under the Kigali Amendment, it is necessary to continuously monitor and identify long-term HFC emission trends in Eastern Asia", we have mentioned the classifications in parentheses directly after 'eastern Asia' for clarity: (non-Article 5 earlier starts: Japan; Article 5 Group 1 countries: China, South Korea, North Korea, and Taiwan).

(3) L87: Change 'in eastern Asia' to 'in these four countries' since the dearth of recent research doesn't seem to apply to China (L82-84).

→ The reviewer made a good point. Regarding the need for research in Eastern Asia, we mentioned that there was no estimate of emissions for a sufficiently long-term period before and after the Kigali amendments based on high-frequency regional background observations, and revised it to clarify as follow:

"However, since then, no recent research has been conducted over a sufficiently long period before and after the Kigali Amendment to investigate HFC emissions in Eastern Asia based on high-frequency regional background observations. Hence, there has been a lack of recent research on HFCs, culminating in a dearth of information on current HFCs emission trends in eastern Asia."

Research in China also has predominantly been conducted in specific regions or over short periods. Pu et. al. (2020) estimated emissions using the inter species correlation (ISC) method based on 3L canister sampling during the period 2012-2016 for the Yangtze River Delta. Zeng et al. (2020) sampled with 2L canisters in the Greater Pearl River Delta region and estimated emissions using the ISC method. Observations were made from 2001 to 2016 at several field locations, but observations were made over a short period of time like campaign, rather than continuous observations at the same location. Ding et al. (2023) and Yi et al. (2023) estimated bottom-up emissions of HFC-134a for North China Plain and 4 big cities, respectively. The last comprehensive top-down research representing

national/regional of China was done by Yao et al. (2019), covering data only up until 2017. Therefore, there is a continuous need for national/regional-scale emission estimates for China as well. Given this, it seems more appropriate to refer to the group collectively as 'eastern Asia' rather than 'four countries'.

L106: Measurements are taken every 2 hours. How long is the sample duration?

→ The sampling duration is 20 minutes. We mentioned in the manuscript.

L107: State how many samples were collected from 2008-2020 and used here. Around 50,000?

→ The number of samples used in this study from 2008 to 2020 is approximately 26,000 for each HFC substance. We mentioned in the manuscript.

L154: Define J(p) within the text. Define T.

→ Revised. We presented the cost function J(p) in the sentence, and added T, which means for transpose of matrix, to the manuscript.

L169: Describe the working tank.

→ The working tank filled with clean air on-site at Gosan station on a clear, rainless day. It is utilized for drift correction of the detector. We added this sentence to the manuscript.

L183: Describe what's meant by flattened.

→ Flattened means that each grid cell was evenly distributed to have the same value for a specific area (each country or eastern China). We added this sentence to the manuscript.

L195: Why observation sites (plural) if Gosan is the only station being used? Do you mean the same subset of data was used for both inversion models?

→ Yes, this study (both inversion frameworks) used the same Gosan observation data. To prevent any misinterpretation regarding the input data, we've removed the phrase of "To exclude differences arising from the variability of observation sites"

L249: HFC-134a emissions from North Korea and Taiwan are up to 1-2 Gg/yr (Figure S2) or 10-20% of values from eastern China and Japan, which isn't nearly negligible. The average Taiwan emission peak is actually fairly similar to South Korea (Table 3). Use different wording and add a brief discussion.

→ Revised. We briefly mentioned the emissions of North Korea and Taiwan. Taiwan's emission account 60% of South Korea's emissions by the late 2010s.

The sentence, "Due to the nearly negligible emissions from North Korea and Taiwan, these regions are not discussed", has been deleted and replaced with the following sentence: "North Korea and Taiwan account for 5 to 10% of the emissions from eastern China and Japan, with Taiwan's emissions particularly increasing to reach 60% of South Korea's emissions by the late 2010s."

L286: Add a sentence or two describing South Korea (Figure 5c). The inversions agree well but seem to greatly exceed the inventory? Please briefly discuss North Korea and Taiwan, as you do for HFC-125.

→ Following the reviewer's comments, we mentioned the increasing trend in Korea. The gaps with Korea's inventory were also discussed in the conclusion. The reason for the very low reported (or unreported) emissions of HFC-32, HFC-125, and HFC-143a in South Korea is that the national inventory compilation does not yet account for the amount of HFCs contained in blended refrigerants such as R-410A and R-404A. Additionally, we mentioned that emissions from North Korea and Taiwan were very minimal up to 2012, but began to increase after 2013, showing a trend similar to that of HFC-125.

The written sentence is: "South Korea have steadily increased from 0.2 Gg/yr to 1.6 Gg/yr from 2008 to 2020, an eight-fold increase. North Korea and Taiwan emissions were less than 0.5 Gg/yr for the overall period. From 2008 to 2012, emissions were almost negligible,

close to zero, but began to gradually increase starting in 2013. This trend is similar to that of HFC-125."

L338: The statement that South Korea showed the most substantial HFC-152a emissions growth doesn't seem correct. Japan's emissions increased from 0.8 to 3.4 Gg/yr (2.6 Gg/yr increase, L333) while South Korea increased from 0.06 to 1.4 Gg/yr (1.3 Gg/yr increase).

→  Yes, it seems it was indeed a phrase that could cause misunderstandings. While Japan has seen a larger absolute increase, we meant to indicate that Korea has the highest relative growth rate (more than 20 times). Therefore, to avoid confusion, we have revised the term from 'growth' to 'growth rate', as follows: "South Korea had highest emissions growth rate of HFC-152a among all countries."

L354: Why is 2015 omitted? I wondered this throughout the paper.

→  As HFC emissions were in an increasing trend consistently across all compounds for 2015, we excluded 2015 as a transitional year and focused on comparing the spatial distributions of HFCs before 2015 to those after 2015.

L376: Add error bars to the emissions, which are given to 0.1 Gg/yr.

→  Revised. Uncertainty range has been added.

L376: The text states a peak of 116 $CO_2$-eq Tg/yr in 2018, but Figure 10 shows a peak of around 128 $CO_2$-eq Tg/yr in 2018. The 2020 value of 110 Tg/yr (L375) also seems low. Please check.

→  Thanks for pointing this out. The CO2-eq HFC emissions indicated in the text was written as the value calculated as previous GWP. The exact values corrected according to the GWP from Liang and Rigby et al. (2022). Fig. 10 shows Liang and Rigby et al. (2022)'s calculated GWP is correct.

L408: Add 'except North Korea' since it didn't have the largest portion of HFC-134a (Figure 11).

→ Revised.

L414: The changes to HFC-134a and HFC-125 in eastern China are 9-10%, which is still substantial.

→ Yes, we agree with the reviewer's opinion. Eastern China has seen relatively smaller changes compared to Korea and Japan, with the changes in HFC-134 and HFC-125 being about 10%. Following the reviewer's comments, we have revised the sentence to: 'In contrast, in eastern China, the changes in the percentage of each HFC over the years have been relatively small, though the changes to HFC-134a and HFC-125 are still substantial at about 10%.'"

L418: This can read as if HFC-134a and HFC-125 are the environmentally friendly alternatives rather than the gases to be reduced.

-> Revised. We have revised the confusing sentence to clearly state. We have split the original long sentence into two sentences as follows: "The changes in the proportion of HFCs can be reflected by each country's industrial structure and policies. Based on this, it may be effective for each country to develop and transition to environmentally friendly alternatives, specifically targeting the reduction of HFC-125 and HFC-134a, to respond to global warming"

L428: 'on the other hand' doesn't apply if both have increased steadily (L427 and L429). Change the second increased steadily to 'increased more slowly'.

→ Revised.

L472: The text here states an accelerating trend since 2015 while L30 states 2016. Add that emissions have subsequently slowed or decreased since 2019.

→ The part in the abstract described as 'since 2016' has been replaced to 'from 2016 to 2018'. Instead of 'since 2015', we have changed the expression to 'after 2015' and revised the sentence to: 'Emissions of HFCs have an accelerating upward trend after 2015 and have subsequently decreased for 2019-2020.'"

L482: Figure 12b shows a peak of ~15% in 2018 but smaller values at other times (L437), so using 15% as the post-2015 value seems too high both here and in the abstract.

→  Revised. The value of 15% has been revised to 13%, the average for 2016-2020. This is the same as an error in abstract.

Increase the font size in the Figures. Figure 9 is especially hard to read, even blown up.

→ Revised. In Figure 9, the numerous images were arranged horizontally, so it was difficult to look. Therefore, by dividing the substance and changing the orientation to vertical, all images are now allocated to one page, facilitating easier verification of each image.

In each Figure label the panels (a, b, c...) and define them in the caption.

→ Revised. We indicated labels such as (a), (b)... in all figures and explained in the captions.

Figure 10: Label the panels and delete 'respectively'.

→ Revised.

(4) Figure 12: Define all terms (Twn as Taiwan...). The caption for (b) only refers to half the legend (eastern Asia to global total) so make sure to also define 'non-Annex-1...to unreported'.

→ Revised.

Supplement:

L49: State in the Figure S8 caption that the proportions are based on mass percentage.

→ Revised.

L51: Delete 'except North Korea' since HFC-134a also accounts for the largest percentage in North Korea (53%). (It seems like the interpretation of Figure 11 and Figure S8 was reversed.)

→ Revised.

L52: HFC-125 increased by 10% (from 15 to 25%) in eastern China and HFC-152a decreased by 12% (23 to 11%), which is more than 'slight'.

→ Revised. The substances that show almost no change are HFC-134a and HFC-143a, and we have removed the term 'slight.' The sentence has been revised to: 'Eastern China shows little change in the proportion of HFC-134a and HFC-143a over time (including increases in HFC-32 and HFC-125 and decreases in HFC-152a)

Figure S1: The scale is difficult to read.

→ Revised.

Table S1: State the year of the prior emissions (2008).

→ Revised.

**Response to Reviewer #2**

We appreciate your very meaningful comments.

It gave us a deeper understanding of what we overlooked and didn't take into account, which enriched the manuscript.

Major comments:

Section 3.2. Can I make a structure change for the authors to consider here? I see that discussions in this section is following a compound by compound flow, similar to Chapter 2 in WMO2022. To me, this approach might be amiss in terms of the key conclusions you can draw from this study. The emission trends shown in Figure 3 for all HFCs, except 143a, point to two distinctive period 2016-2018 during which the two dominant emitting regions Eastern China and Japan show rapid increase in emissions and 2019-2020 during which emissions either decreased or plateaued. Consider add a summary paragraph here right at the beginning of Section 3.2 (where Figure 3 sits), describing trends of all gases together. And later, when you discuss each compound, you can refer to the corresponding discussion. Even though reasons for this notable change in 2019 might be different for different countries and not necessary directly tied to F-gas regulation, but in my view such a trend is note-worthy and should be pointed out clearly at the beginning of discussion, instead of being buried among many other details in each compound section.

→ We think this is a very good opinion. It would be good to have an introduction that briefly summarizes the overall trends of HFCs. So, we have written a brief summary of HFCs emission trends at the beginning of section 3.2, which describes inversion results. Afterwards, we move on to the analysis of individual HFCs.

The written paragraph is as follows:

The HFC emissions estimate from 2008 to 2020 in Eastern Asia reveals distinct phases of changes. HFC emissions gradually increased or remained relatively constant until 2015. However, between 2016 and 2018, there was a sharp increase in emissions, especially evident in eastern China and Japan, primarily affected by HFC-134a, HFC-32, and HFC-125.

Since 2019, the HFC emission trend in Japan has shifted towards stabilization or reduction, suggesting the onset of more stringent regulatory impacts. Contrastingly, although the emissions in South Korea, North Korea, and Taiwan are lower compared to China and Japan, there has been an increase over the years, with a noticeable acceleration in South Korea's emissions.

Figure 10 and Section 3.4 provide compelling evidence that emission trends for 2019-2020 are distinctively different from previous time period. This should be a key message from your paper! Currently the discussion in section 3.4 focuses on the details of emissions from each country, the relative contribution from each HFC, etc. In a way, you are look at each individual tree but forgot about the forest, that the cumulative emissions of five most abundant HFCs from East Asian countries made a turn-around point in 2019 (whether it is due to the Kigali Amendment influence or not).

→ Yes, the decreasing trend since 2019 is the effect of Japan's reduced emissions. This may be related to F-gas regulation and may have important implications. The title of the second category summarized in the Conclusion section has been revised to "Unexpected increase in HFC emissions late 2010s and a recent decline in Japan" to further emphasize the turning point in Japan. The decrease in Japan post-2019 requires ongoing monitoring and understanding. After 2019, while there has been a reduction in emissions in East Asia due to Japan's influence, the reduction is modest and the levels of HFC emissions are still higher than before the rapid increase.

I would think that Figure 10 will be a key take-away figure when this paper is published. I have a few cosmetic suggestions to make this figure more appealing to eyes. The individual stacked-up bars for each year leave a very discrete view. You may want to make the bars wider (leaving only fine space between bars) so that they provide a more continued temporal flow from one year to the next. Second, the current choices of colors are not optimal. There is enough contrast, but colors are clashing to eyes. Third, the legend for countries or HFCs are flipped in order when compared with the sequence they are stacked in the figure.

→ Revised. By increasing the width of the bar in the image, the continuity of the data was highlighted a little more. The colors have been toned down, and the legend has been modified to match the order in which the data is stacked.

Section 4 - the major findings from this paper, I would suggest that "reverse of increasing emissions trend from 2008-2018 to a decreasing trend in 2019-2020" need to be added as one major finding. In a way, this is a more significant finding than all the three findings you listed.

→ We think this is related to the previous comments. As we said above, one interesting finding in this study was the rapid increase in eastern Asia. We absolutely agree that the decline since 2019 is something we should pay attention to. However, despite a turning-point since 2019, emissions are still higher than before the sharp increase. It is clear that the decline trend since 2019 is something we need to continue to monitor and confirm. Therefore, we have emphasized it by mentioning it at subtitle 2 in Conclusion section "Unexpected increase in HFC emissions late 2010s and a recent decline in Japan".

Minor comments:

L20 & L96. It should be Kigali Amendment (upper case A).

→ Revised.

I would suggest call them "most abundant HFCs" instead of "major HFCs", here and after, to be consistent with WMO 2022. What exactly is a major HFC anyway?

→ Revised. The expression major seems to be a somewhat ambiguous expression. According to the reviewer's opinion, it was changed to the most abundant HFCs.

"Our most important finding is that HFC emissions in eastern China and Japan have sharply increased since 2016." Looking at figure 3, this is not an accurate statement as the increase for Japan and China (except HFC-143a) only lasts from 2016 to 2018. Emissions for HFC-32, 125, 152a in both regions have decreased since 2019.

→ Revised. The expression since 2016 does not seem to express the overall trend. Since the emissions of HFCs have decreased since 2019, the expression has been clearly modified to say from 2016 to 2018, which indicates a noticeable increase, instead of since 2016.

"HFCs are not ODSs since they are oxidised more readily in the troposphere and do not contain chlorine atoms." Being oxidized in the troposphere is not the reason why they are not ODSs. A lot of the ODSs, e.g. CH3Br, CH3Cl, CH3CCl3, are primarily oxidized in the troposphere and have comparable or shorter lifetimes than many HFCs.  Suggest change to "HFCs do not contain ozone-depleting chlorine atoms".

→ Revised. According to the reviewer's opinion, oxidized in the troposphere... The phrase was deleted and the sentence was modified to clarify: "HFCs do not contain ozone-depleting chlorine atoms."

L51 & L95. Paris agreement ◊ Paris Agreement

→ Revised.

It should be Rigby et al. (2014). Parenthesis are missing.

→ Revised.

Define AGAGE here.

→ Revised. AGAGE, which was defined in the Instrumentation section, was defined in the Introduction, which was mentioned first.

May be rephrase to "Previous regional studies demonstrated that there is reasonable consistency between bottom-up and top-down emission estimates Europe (Graziosi et al., 2017) and the United States (Hu et al., 2017)"

→ Revised.

"(mole fractions)", shouldn't you add the definition at L99, instead of here?

→ Revised.

Can you explain briefly what is "a 2-year assimilation time window"? To me this is a bit jargon-ish, not necessary clear to readers what do you mean by "assimilation".

→ Yes, the 2-year assimilation time window means that the temporal matrix of the observation is constructed for 2 years of observation data during the inversion process. In other words, it means that posterior emission is estimated using two years of observation data observed in Gosan. By deriving results from moving windows per year, a smoother annual change in emissions can be obtained. Therefore, we have clarified the sentense as follows: "In this study, for both inversion frameworks we chose to use a 2-year temporal matrix of the observation data as an assimilation time window, which increased the sensitivity through a more extended temporal window than compared to a 1-yr window (which is commonly used for synthetic gases), and to resolve the emissions with 1-year resolution."

Change to "presented in this study".

→ Revised. "Presented in the results of" was changed to "presented in this study."

L142 & L198. Missing parenthesis in references for "2021", "2018".

→ Revised. For reference notation, Kim et al. (2018), Arnold et al. (2018), Manning et al. (2021) was modified by adding parentheses.

It is commonly referred to as "Asian Summer Monsoon", not "summer Asian monsoon". Consider revise.

→ Revised. We modified it with Asian Summer Monsoon.

Figure 2. Nice way of showing pollution magnitudes.

→ Thank you. The annual enhancement plot is a way to help visually perceive the pollution magnitude.

Figure 4 caption. Can you clarify what is "average values"? Average of the FLEXPART and NAME results?

→ It refers to the average of the inverse estimation model results of FLEXPART-FLEXINVERT+ and NAME-InTEM. The meaning was clarified by specifying 'and average values of both inversion framework results' in the sentence.

L255-257. Here, are you commenting on the small differences between the UNFCC inventory vs. the EDGAR-v7 inventory? I found this sentence to be somewhat out of place in this paragraph, as the remaining paragraph mostly discuss the gap between top-down and bottom-up inventories. A good and punchy way of starting a paragraph is to write a key summary sentence that summarize of the main finding of the paragraph, e.g. "There are significant and growing gaps between the top-down emission estimates and bottom-up estimates in eastern China and Japan, etc.". You can note the small differences between the UNFCC inventory vs. the EDGAR-v7 inventory somewhere in the paragraph instead of mentioning at the beginning of the paragraph.

→ Thank you for comments. According to your suggestion, we wrote "There are significant and growing gaps between the top-down emission estimates and bottom-up estimates in eastern China, Japan, and South Korea" at the start of the paragraph. Further, we mentioned the difference briefly between UNFCCC reported inventory and EDGAR as follows:

"UNFCCC reports are based on national statistical data that follow Intergovernmental Panel on Climate Change (IPCC) methodologies, while EDGAR uses a uniform methodology to provide globally consistent gridded estimates from various comprehensive datasets. Therefore, this may lead to discrepancies due to different assumptions and data sources."

L277-278. Consider revise to: HFC-32 abundance has been increasing substantially in recent years.

→ Revised. We modified the sentence "HFC-32 is one of the substances with an abundance that has been increasing substantially in recent years" to "HFC-32 abundance has been increasing substantially in recent years" based on the reviewer's comment.

L279-282, why do you use eastern China sometimes and Eastern China other times? Please be consistent.

→ Capitalize only the subject at the beginning of the sentence, Eastern China, and keep the rest consistently written as eastern China.

L304-306. This is an important finding about HFC-125. I would suggest you move this sentence to the beginning, right after HFC-125 usages.

→ Revised.

"HFC-152a, which has the lowest GWP (148) among HFCs" This is not a correct statement. There are many HFCs that have lower GWP. Did you mean among KA-regulated HFCs or the five most abundant HFCs in this study?

→ We agree that that was a misleading expression. That was the meaning of among HFCs in this study. We added 'in this study'.

L353-354. Why is this sentence a paragraph of its own? In fact, you don't need this sentence at all. It is just a figure description. You can delete it and add "(Figure 9)" at the end of the sentence on L355.

→ Revised.

Change to "HFCs are potent anthropogenic greenhouse gases". You don't need "," here.

→ Revised.

Change "," after growth to "." The second part is a sentence on its own.

→ Revised.

You probably should add "South Korea" after Gosan to recapture things again for those who only read abstracts and conclusions.

→ Revised.

"are" is missing after there.

→ Revised.